# Circadian neurons in the paraventricular nucleus entrain and sustain daily rhythms in glucocorticoids

Jeff R. Jones [1,2], Sneha Chaturvedi [1], Daniel Granados-Fuentes[1] & Erik D. Herzog [1✉]

Signals from the central circadian pacemaker, the suprachiasmatic nucleus (SCN), must be decoded to generate daily rhythms in hormone release. Here, we hypothesized that the SCN entrains rhythms in the paraventricular nucleus (PVN) to time the daily release of corticosterone. In vivo recording revealed a critical circuit from SCN vasoactive intestinal peptide (SCN[VIP])-producing neurons to PVN corticotropin-releasing hormone (PVN[CRH])-producing neurons. PVN[CRH] neurons peak in clock gene expression around midday and in calcium activity about three hours later. Loss of the clock gene *Bmal1* in CRH neurons results in arrhythmic PVN[CRH] calcium activity and dramatically reduces the amplitude and precision of daily corticosterone release. SCN[VIP] activation reduces (and inactivation increases) corticosterone release and PVN[CRH] calcium activity, and daily SCN[VIP] activation entrains PVN clock gene rhythms by inhibiting PVN[CRH] neurons. We conclude that daily corticosterone release depends on coordinated clock gene and neuronal activity rhythms in both SCN[VIP] and PVN[CRH] neurons.

---

[1] Department of Biology, Washington University, St. Louis, St. Louis, MO, USA. [2] Present address: Department of Biology, Texas A&M University, College Station, College Station, TX, USA. ✉email: herzog@wustl.edu

Normal behavior and physiology depend on the circadian, or near-24 h, the release of hormones at the optimal time of day. Disruption of these rhythms due to disease or lifestyle is associated with numerous pathologies including affective and metabolic disorders[1,2]. Understanding the neural circuits regulating the daily pattern of hormone production is, therefore, a central problem in neuroscience. The daily timing of hormone release is controlled by a central circadian pacemaker, the suprachiasmatic nucleus (SCN). SCN neurons contain a daily transcriptional-translational negative feedback loop consisting of core clock genes and proteins (e.g., *Per1/2*, *Cry1/2*, *Bmal1*, *Clock*) and exhibit daily rhythms in neuronal activity that communicate circadian time to the rest of the brain and body[3,4]. The SCN is necessary for daily rhythms in hormone production, as each of these rhythms are lost after SCN ablation[5–7]. SCN transplants can restore locomotor, but not endocrine, rhythms, indicating that synaptic connections from the SCN are required for daily rhythms in hormone release[8]. However, fundamentally, we do not know how the SCN regulates daily rhythms in different hormones that each peak at different times of the day. For instance, while melatonin peaks at night in all vertebrates, glucocorticoids peak just prior to waking in both diurnal and nocturnal species[9,10]. Consequently, it is important to investigate how signals from the SCN are decoded to generate circadian rhythms in hormone release.

The release of glucocorticoids, including cortisol in humans and corticosterone in rodents, is one of the highest amplitude circadian outputs and thus provides an excellent model system to understand the daily regulation of hormone release. Glucocorticoids are released each day just prior to waking to prepare the brain and body for physical and cognitive activity and their rhythmic release is essential for human health[11–14]. Normal cortisol rhythms are disrupted in Cushing's syndrome, in Addison's disease, in most patients undergoing glucocorticoid therapy, in patients with chronic stress or post-traumatic stress disorder, and in patients with anxiety or depression, which results in numerous physiological and psychiatric symptoms[15–20]. Similarly, cortisol rhythms and sleep schedules can become misaligned in shift workers and in social jet lag, which contributes to an increased incidence of disease[21–24]. Despite its clear importance to cognitive and mental health, the mechanisms underlying the circadian release of glucocorticoids are unclear.

Glucocorticoids are ultimately released as part of the hypothalamic-pituitary-adrenal (HPA) axis. The HPA axis regulates the body's response to stress through corticotropin-releasing hormone (CRH)-producing neurons in the paraventricular nucleus of the hypothalamus (PVN), which promote the release of adrenocorticotropin hormone (ACTH) from the pituitary to induce the adrenal glands to produce glucocorticoids. This stress-induced activation of the HPA axis is superimposed on a baseline circadian release of glucocorticoids that is ultimately regulated by signals from the SCN to PVN$^{CRH}$ neurons that act to induce or suppress the release of ACTH[25]. Additionally, the SCN acts synergistically on a distinct population of pre-autonomic neurons in the PVN to determine the times at which the adrenals are most sensitive to ACTH[26,27]. Importantly, we do not know exactly how SCN neuronal input affects PVN neurons to appropriately time glucocorticoid production. Previous studies have shown that the release of arginine vasopressin (AVP) from the SCN inhibits corticosterone release in rodents[28–30]. However, as the SCN is a heterogeneous network[31,32], we lack a mechanistic understanding of what specific cell types in the SCN promote the peak and suppress the trough of the circadian corticosterone rhythm.

SCN neurons that release the neuropeptide vasoactive intestinal peptide (VIP) have been implicated in sustaining daily rhythms in the SCN and glucocorticoid release[33–38]. For instance, mice genetically deficient for *Vip* or *Vipr2* (which encodes the VIP receptor VPAC2) exhibit disrupted corticosterone rhythms[39,40]. VIP injected into the PVN 2 h after light onset can induce ACTH and corticosterone release within 15 min[41]. Similarly, deleting SCN$^{VIP}$ neurons blunts corticosterone rhythms[42]. Although these results suggest a stimulatory role for VIP, chemogenetically inhibiting SCN$^{VIP}$ neurons during the midday can transiently increase corticosterone release[43]. Thus, the roles of SCN$^{VIP}$ neurons and VIP in coordinating daily rhythms in the SCN and its outputs are unclear.

PVN cells exhibit circadian rhythms in clock gene expression and in firing rate[44–46]. Among these, PVN$^{CRH}$ neurons exhibit circadian rhythms in *Crh* mRNA and CRH levels vary with time of day in cerebrospinal fluid[47–49]. Although >75% of PVN$^{CRH}$ neurons express the core clock gene *Per1*, it is not known if PVN$^{CRH}$ neurons express daily rhythms or if their rhythmicity is necessary for circadian glucocorticoid release. We, therefore, hypothesized that circadian signals from SCN$^{VIP}$ neurons to rhythmic PVN$^{CRH}$ neurons are necessary for daily glucocorticoid rhythms.

Here, we find a critical circuit from SCN$^{VIP}$ neurons to PVN$^{CRH}$ neurons that times the daily release of corticosterone to around waking. We find that in a light cycle, PVN$^{CRH}$ neurons recorded in vivo from freely-behaving mice over several days peak in *Period2* clock gene expression around midday and peak in calcium activity a few hours later. These rhythms persist in constant darkness and dampen in constant light. The loss of the core clock gene *Bmal1* in CRH neurons results in arrhythmic PVN$^{CRH}$ neuron calcium activity and dramatically reduces the amplitude and precision of the daily rhythm in corticosterone release. Discrete activation or inactivation of SCN$^{VIP}$ neuron activity reliably alters the peak amplitude of corticosterone release and PVN$^{CRH}$ neuron calcium activity, and daily SCN$^{VIP}$ activation entrains PVN clock gene rhythms by inhibiting PVN$^{CRH}$ neurons. Together, these results demonstrate that the daily corticosterone surge depends on coordinated clock gene expression and neuronal activity rhythms in both SCN$^{VIP}$ and PVN$^{CRH}$ neurons.

## Results

**PVN$^{CRH}$ neurons exhibit synchronous daily rhythms in clock gene expression and neuronal activity**. To determine whether the physiology of PVN$^{CRH}$ neurons changes with time of day, we developed methods to monitor clock gene expression and calcium activity in PVN$^{CRH}$ neurons in freely-behaving mice longitudinally and at high temporal resolution. We used in vivo fiber photometry to record cell-type-specific Venus fluorescence as a readout of *Per2* transcriptional activity in the PVN of heterozygous *Crh*-IRES-Cre knock-in mice injected with a Cre-dependent virus encoding the fluorescent clock gene reporter *Per2.DIO.Venus*[50] (Fig. 1a, b). We measured *Per2* fluorescence over 60 s every 15 min for multiple days in mice implanted with a fiber optic cannula above their PVN (Supplementary Fig. 1a). We found daily rhythms in *Per2* in PVN$^{CRH}$ neurons in constant light (LL), constant darkness (DD), and in a 12:12 light/dark cycle (LD), peaking during the daily minimum of locomotor activity (Supplementary Fig. 1b). The period of *Per2* expression rhythms in PVN$^{CRH}$ neurons depended on ambient light, with a period of ~24 h in LD and shorter and longer in DD and LL, respectively (Fig. 1c, d). Although all mice showed daily rhythms in PVN$^{CRH}$ *Per2* expression in all lighting conditions, LL significantly reduced rhythm amplitudes (Fig. 1e, f). Importantly, control mice did not show daily rhythms in fluorescence (Supplementary Fig. 1c). We found that clock gene rhythms in PVN$^{CRH}$ neurons peaked at midday in LD (ZT 5.9) and mid-subjective day in DD (CT 5.3;

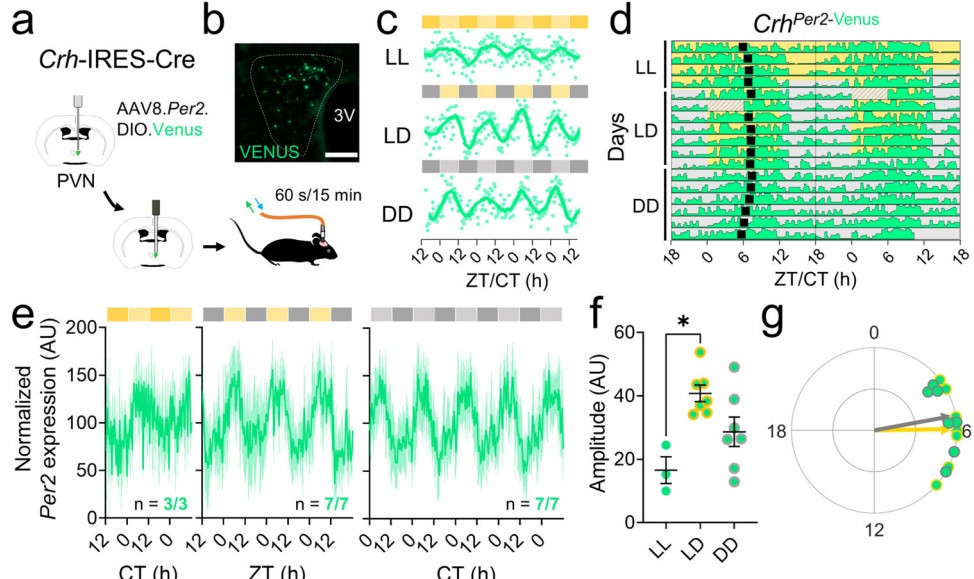

**Fig. 1 PVN^CRH neurons exhibit daily rhythms in clock gene expression that peak at midday. a** Schematic of in vivo fiber photometry recording of *Per2* expression from targeted cells in the hypothalamic paraventricular nucleus (PVN) of *Crh*^Cre/+ mice. **b** Representative (*n* = 4 replicates) PVN (dashed line) image depicting Cre-dependent Venus expression (light green). 3 V, third ventricle. Scale bar = 100 μm. **c** Representative in vivo *Per2*-Venus fluorescence from PVN^CRH neurons averaged over 60 s every 15 min (light green dots) for 4 days from mice housed in constant light (LL), a 12 h:12 h light:dark cycle (LD), or constant darkness (DD). Data were smoothed with a 4 h Savitzky–Golay filter (light green line). ZT zeitgeber time; CT circadian time. **d** Representative double-plotted actogram of *Per2*-Venus fluorescence (light green) recorded from PVN^CRH neurons in a mouse over 17 days in LL, LD, and DD. Black squares depict daily *Per2* acrophases. Recording was interrupted for part of day 6 (hatched rectangle). **e** Normalized levels of *Per2* expression from PVN^CRH neurons in mice recorded in LL (dark yellow and yellow bars; *n* = 3/3 mice rhythmic, JTK cycle, *p* < 0.020 or less, see Supplementary Table 1 for *p*-values of individual mice), LD (gray and yellow bars; *n* = 7/7 mice rhythmic, JTK cycle, *p* < 0.001 for all mice), and DD (gray and light gray bars; *n* = 7/7 mice rhythmic, JTK cycle, *p* < 0.001 for all mice). Light green lines and shading depict mean ± SEM. **f** The peak-to-trough amplitudes of PVN^CRH *Per2* rhythms in LL, LD, and DD. *n* = 3, 7, 7 mice. Brown–Forsythe ANOVA with post-hoc Dunnett's multiple comparison's test, *p* = 0.007. Lines depict mean ± SEM. **g** Rayleigh plots of *Per2* expression in PVN^CRH neurons from mice housed in LD (light green dots with yellow outlines; peak time ZT 5.9, Rayleigh test, *p* = 0.001) and in DD (gray outlines; peak time CT 5.3, Rayleigh test, *p* = 0.001). In this and other Rayleigh plots, arrows point to the mean time of day when the rhythm peaked, and the length of the arrow indicates variability in the data ranging from 0 (peaked at random times) to 1 (all recordings peaked at the same time). Source data are provided as a Source Data file.

Fig. 1g) when measured with *Per2*-Venus or with a biolumines-cent reporter of *Per2* (Supplementary Fig. 1d). Together, these results demonstrate that PVN^CRH neurons exhibit sustained circadian rhythms in gene expression in vivo that synchronize to the light cycle.

To determine whether the daily rhythm in gene expression relates to neuronal activity, we next used in vivo fiber photometry to monitor calcium fluorescence from PVN^CRH neurons in heterozygous *Crh*-IRES-Cre knock-in mice injected with a Cre-dependent virus encoding the genetically encoded calcium reporter GCaMP6s[51] (Fig. 2a, b). We measured calcium events and levels in mice implanted with a fiber optic cannula above their PVN for 10 min per hour for multiple days by simultaneously exciting GCaMP6s with its calcium-dependent excitation wavelength and its calcium-independent isosbestic excitation wavelength (Supplementary Fig. 2a). PVN^CRH neu-rons responded to a stressor with a large increase in calcium fluorescence, consistent with prior reports[52,53] (Supplementary Fig. 2b). PVN^CRH neurons showed significant daily rhythms in calcium event frequency in all mice in LD and DD, peaking in the mid-afternoon (ZT 7.1, CT 7.7), about 2 h after the peak time of their *Per2* expression (Fig. 2c–e). These rhythms persisted in LD and DD for as long as we recorded (>10 d; Supplementary Fig. 2c) and paralleled daily rhythms in integrated calcium levels that also peaked around mid-afternoon (ZT 7.0, CT 8.0; Supplementary Fig. 2d, e). Control *Crh*-IRES-Cre mice injected with a Cre-dependent EGFP virus showed no daily rhythms in fluorescence (Supplementary Fig. 2f–h).

To evaluate whether these circadian rhythms were intrinsic to the PVN, we next recorded GCaMP6s calcium fluorescence every hour from individual CRH neurons in ex vivo PVN slices. Similar to in vivo recordings, calcium fluorescence of individual CRH neurons in vitro peaked in the mid-subjective afternoon (CT 8.5; Supplementary Fig. 2i–k). However, unlike in vivo PVN^CRH calcium rhythms, ex vivo calcium rhythms did not persist longer than ~36 h, suggesting that CRH neurons require extra-PVN input for sustained daily rhythms.

**Loss of BMAL1 in CRH neurons eliminates the reliable, high-amplitude daily rhythm of corticosterone.** To determine if the CRH neuron circadian clock mediates daily rhythms in corticosterone release, we generated a line of *Crh*^Cre/Cre; *Bmal1*^fl/fl mice (KO) in which the core clock gene, *Bmal1* or *Arntl*[54], was ablated in CRH neurons, but intact in the SCN and elsewhere in the PVN (Fig. 3a and Supplementary Fig. 3a, b). We found that two copies of *Crh*-Cre were required to completely ablate BMAL1, consistent with several other studies with floxed *Bmal1* mice[38,55]. *Crh*^Cre/Cre homozygotes showed no physiological (e.g., corticosterone rhythmicity), behavioral (e.g., locomotor activity), or anatomical (e.g., number of CRH⁺ neurons) deficits. Similarly, there was no significant difference between the number of CRH-immunolabeled neurons in *Crh*-Cre homozygotes, heterozygotes, or *Crh*^Cre/Cre; *Bmal1*^fl/fl mice (Supplementary Fig. 3c). Together, these data demonstrate that the presence of homozygous *Crh*-Cre and the knockout of *Bmal1* in CRH neurons each has no effect on PVN^CRH expression.

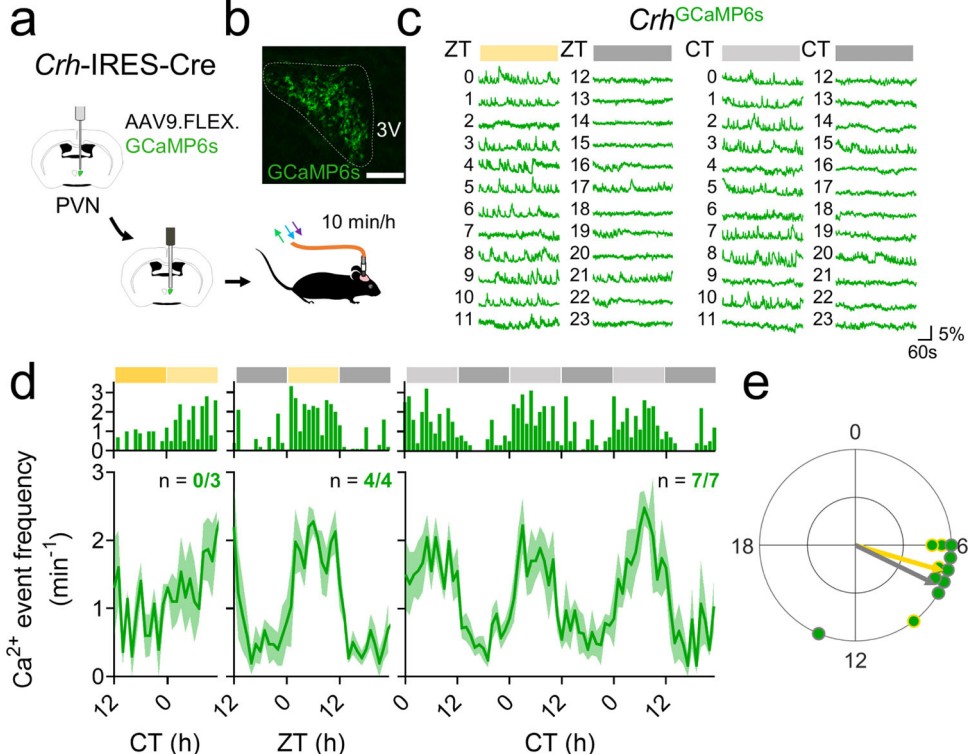

**Fig. 2 PVN^CRH neurons exhibit daily rhythms in calcium activity that peak at mid-afternoon. a** Schematic for in vivo fiber photometry recording of calcium activity in the PVN of $Crh^{Cre/+}$ mice. **b** Representative ($n = 4$ replicates) PVN (dashed line) image depicting Cre-dependent GCaMP6s expression (green). 3V third ventricle. Scale bar = 100 μm. **c** Representative GCaMP6s traces (in $\Delta F/F$) from PVN^CRH neurons recorded hourly from a mouse in LD (12 h:12 h light:dark cycle; left plot, where yellow = light phase) and another in DD (constant darkness; right plot, where light gray = subjective day). ZT zeitgeber time; CT circadian time. **d** Calcium event frequency rhythms from PVN^CRH neurons in mice recorded in LL (constant light; $n = 0/3$ mice rhythmic, JTK cycle, $p > 0.050$ for all mice, see Supplementary Table 1 for p-values of individual mice), LD ($n = 4/4$ mice rhythmic, JTK cycle, $p < 0.001$ for all mice) and in DD ($n = 7/7$ mice rhythmic, JTK cycle, $p < 0.030$ or less, see Supplementary Table 1 for p-values of individual mice). Top, event frequency rhythms from a representative mouse; bottom, event frequency rhythms averaged from multiple mice. Green lines and shading depict mean ± SEM. **e** Rayleigh plots of calcium event frequency rhythms in PVN^CRH neurons from mice housed in LD (green dots with yellow outlines; peak time ZT 7.1, $p = 0.019$) and in DD (gray outlines; peak time CT 7.7, Rayleigh test, $p = 0.003$). Source data are provided as a Source Data file.

To first determine the physiological consequence of ablating BMAL1 in CRH neurons, we used in vivo fiber photometry to monitor calcium fluorescence from PVN^CRH neurons in KO mice injected with Cre-dependent GCaMP6s (Fig. 3b). BMAL1-ablated PVN^CRH neurons responded to a stressor with a large increase in calcium fluorescence, indicating that the stress response of these neurons remained intact after BMAL1 ablation (Supplementary Fig. 3d, e). In LD, BMAL1-ablated PVN^CRH neurons showed significant daily rhythms in calcium event frequency and integrated calcium levels in all recorded mice, peaking in the mid-afternoon, identical to wild-type PVN^CRH neurons (events ZT 7.3, integrated calcium ZT 8.0; Fig. 3d, e and Supplementary Fig. 3f, g). However, the waveform of BMAL1-ablated PVN^CRH neuron calcium event frequency rhythm changed such that they responded acutely to light but failed to continue firing into the early night (Supplementary Fig. 3h, $p < 0.01$). Remarkably, in DD, BMAL1-ablated PVN^CRH neuron calcium event frequency and integrated calcium levels were each arrhythmic. Together, these results are consistent with light-induced masking and a lack of photoentrainment of BMAL1-ablated PVN^CRH neuron activity.

Next, we customized cages to allow for the non-invasive measurement of fecal corticosterone metabolites from individual mice (Fig. 3b). Mice habituated to the cages in LD for at least 1 week and then in DD for 24 h before we automatically collected fecal pellets every 4 h for three days. Because BMAL1 was present in PVN^CRH neurons of all control genotypes ($Crh^{Cre/+}$; $Bmal1^{fl/fl}$,

$Crh^{Cre/Cre}$, and $Bmal1^{fl/fl}$), and because there was no significant difference in corticosterone rhythmicity among controls, we analyzed all control genotypes together as "wild-type." We found that fecal corticosterone peaked each day around 4 h after subjective dusk in WT mice, but, instead, peaked at unreliable times with low amplitude in KO mice (Fig. 3f and Supplementary Fig. 3i). Mice lacking BMAL1 in CRH neurons had roughly half the average, maximum, and peak-to-trough amplitude levels of fecal corticosterone, with no difference in minimum corticosterone levels compared to WT (Supplementary Fig. 3j). Similarly, the peak-trough amplitude—the ratio between mean corticosterone levels from CT 10-18 (when corticosterone normally peaks) and CT 22-6 (the trough of the corticosterone rhythm)—was significantly lower in KO than in WT mice on all days of collection (Fig. 3g). We found that daily corticosterone peaks were synchronous among individual WT mice, peaking near subjective dusk on each day of collection, but were scattered throughout the subjective day in individual KO mice (Fig. 3h). Accordingly, the standard deviation of peak times of daily corticosterone was more than doubled in KO compared to WT mice on all days of collection (day 1, KO 7.5 h, WT 2.2 h; day 2 KO 5.9 h, WT 2.6 h; day 3 KO 6.4 h, WT 1.1 h). Together, these results demonstrate that the ablation of BMAL1 in CRH neurons results in arrhythmic PVN^CRH calcium activity and reduces the amplitude and precision of daily corticosterone release. We conclude that BMAL1 regulates the cell-autonomous circadian

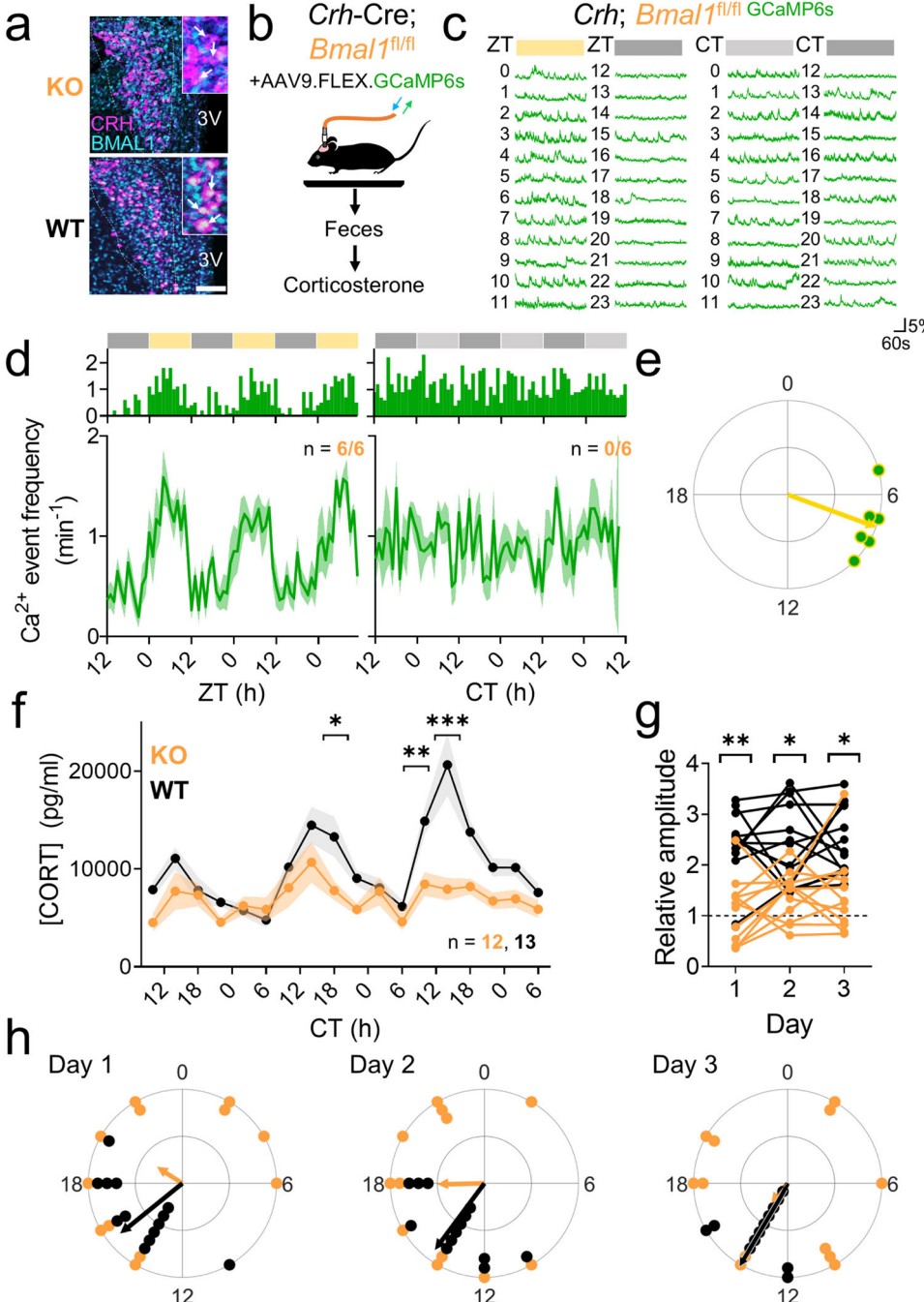

regulation of excitability in PVN[CRH] neurons and their ability to control circadian corticosterone release, but not their response to light or stress.

**The amplitude of circadian corticosterone levels depends on SCN[VIP] neuron activity.** Although these results suggest that the PVN[CRH] clock is necessary for PVN[CRH] neuron calcium activity and corticosterone rhythmicity, it is not sufficient, as PVN[CRH] neurons in KO mice are rhythmic in LD. Similarly, lesioning the SCN or genetically ablating BMAL1 in all SCN neurons eliminates the circadian corticosterone rhythm[7,56]. We, therefore, sought to identify the SCN-to-PVN circuit that is necessary for corticosterone rhythms. To test the hypothesis that firing patterns of SCN[VIP] neurons regulate corticosterone release, we injected the SCN of heterozygous *Vip*-IRES-Cre mice with a Cre-dependent virus encoding the excitatory optogenetic construct

ChR2 or EGFP as a control and subsequently implanted a fiber optic cannula above the SCN[57]. We then tethered each mouse to a fiber optic cable and allowed it to habituate in LD for at least 1 week in a custom cage before feces collection every 4 h for 3 days in DD (Fig. 4a). We chose to optogenetically stimulate mice expressing either ChR2 or EGFP in their SCN[VIP] neurons from CT 11-13, around the time at which corticosterone levels peak and SCN[VIP] neuron activity declines to its daily minimum in WT mice[34,50]. Optogenetic stimulation at this time increased the expression of the immediate-early gene c-FOS (an indicator of neuronal activity) in the SCN of *Vip*-Cre; ChR2, but not *Vip*-Cre; EGFP mice (Fig. 4b), consistent with prior reports[35]. We found that activating SCN[VIP] neurons at subjective dusk acutely and dramatically reduced the peak of the daily rhythm in fecal corticosterone (from 20571 ± 3449 pg/ml in EGFP to 7333 ± 1606 pg/ml in ChR2 mice; Fig. 4c and Supplementary Fig. 4a) and,

**Fig. 3 Ablation of BMAL1 in CRH neurons blunts and desynchronizes circadian rhythms in calcium activity and corticosterone. a** Representative ($n = 4$ replicates) PVN (dashed line) immunohistochemistry images from $Crh^{Cre/Cre}$; $Bmal1^{fl/fl}$ (KO, top) and $Crh^{Cre/+}$; $Bmal1^{fl/fl}$ (WT, bottom) mice depicting CRH (magenta) and BMAL1 (cyan) expression. 3V third ventricle. Scale bar = 100 μm. Inset, higher magnification image. **b** Schematic of fiber photometry and fecal corticosterone collection. **c** Representative GCaMP6s traces (in $\Delta F/F$) from PVN$^{CRH}$ neurons recorded hourly from a KO mouse in LD (12 h:12 h light:dark cycle; left plot, where yellow = light phase) and in DD (constant darkness; right plot, where light gray = subjective day). ZT zeitgeber time, CT circadian time. **d** Calcium event frequency rhythms from PVN$^{CRH}$ neurons in KO mice recorded in LD ($n = 6/6$ mice rhythmic, JTK cycle, $p < 0.040$ or less, see Supplementary Table 1 for $p$-values of individual mice) and in DD ($n = 0/6$ mice rhythmic, JTK cycle, $p > 0.050$ for all mice, see Supplementary Table 1 for $p$-values of individual mice). Top, event frequency rhythms from a representative mouse; bottom, event frequency rhythms averaged from multiple mice. Green lines and shading depict mean ± SEM. **e** Rayleigh plots of calcium event frequency rhythms in PVN$^{CRH}$ neurons from KO mice housed in LD (green dots with yellow outlines; peak time ZT 7.3, Rayleigh test, $p = 0.001$; KO vs. WT peak time Watson–Williams test, $p = 0.796$). Arrhythmic recordings from KO mice housed in DD are not depicted. **f** Corticosterone rhythms over 3 days in WT (black, $n = 13$) and KO mice (orange, $n = 12$). Lines and shading depict mean ± SEM. Mixed-effects model with post-hoc Sidak's multiple comparisons test, *$p = 0.041$, **$p = 0.009$, ***$p < 0.001$. **g** The relative amplitude of the corticosterone rhythm (the ratio of the average level at CT 10-18 divided by the average at CT 22-6) in WT (black) and KO (orange) mice on each day of collection. Two-way repeated-measures ANOVA with post-hoc Sidak's multiple comparisons test, *$p = 0.002$, ***$p < 0.001$. **h** Rayleigh plots of corticosterone rhythms in WT (black dots, peak times Days 1–3 CTs 15.3, 14.4, 14.2, Rayleigh test, $p < 0.001$ on each day) and KO (orange, no significant clustering on any day, Rayleigh test, $p = 0.427, 0.101, 0.720$ on days 1, 2, and 3, respectively) mice on each day of collection. Phases differed significantly between genotypes (two-way circular ANOVA, $p < 0.001$). Source data are provided as a Source Data file.

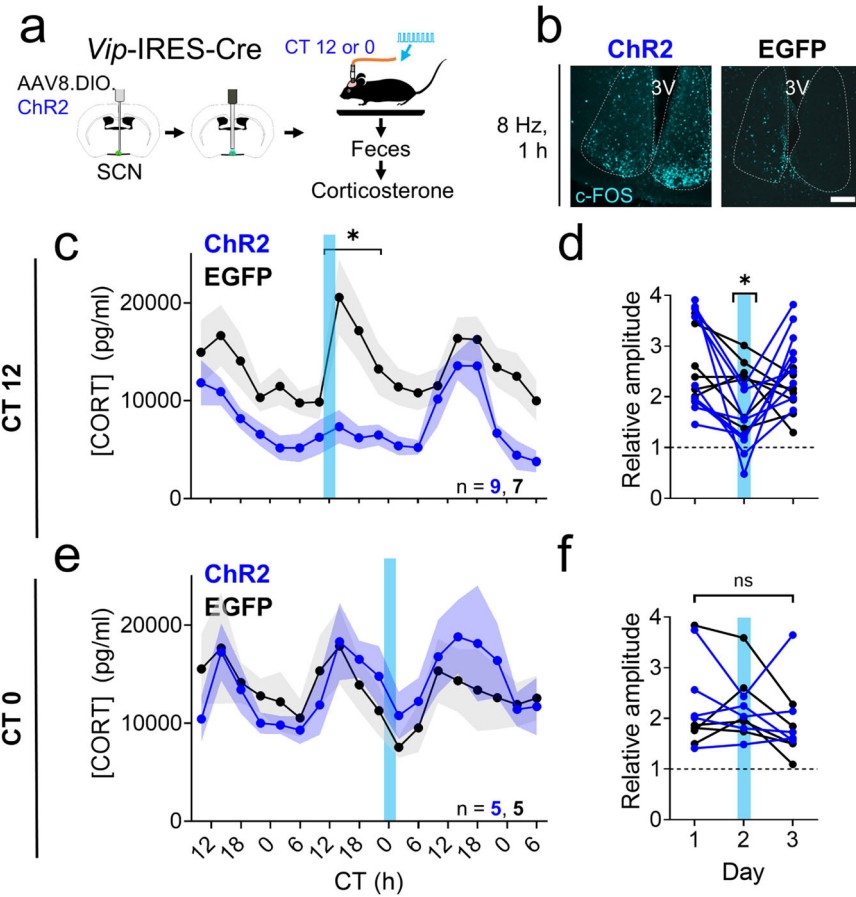

**Fig. 4 Acute activation of SCN$^{VIP}$ neurons blunts circadian rhythms in corticosterone depending on the time of day of stimulation. a** Schematic for concurrent SCN stimulation and fecal corticosterone collection. **b** c-FOS immunoreactivity ($n = 4$ replicates, cyan) after SCN stimulation at CT (circadian time) 12 in $Vip^{Cre/+}$ + ChR2 (ChR2, left) and $Vip^{Cre/+}$ + EGFP (EGFP, right) mice. 3V third ventricle. Scale bar = 100 μm. **c** Corticosterone rhythms over three days in EGFP (black, $n = 7$) and ChR2 mice (blue, $n = 9$). Lines and shading depict mean ± SEM. Blue bar, time of optogenetic stimulation (8 Hz, 10 ms, 470 nm, 2 h from CT 11-13). Mixed-effects model with post-hoc Sidak's multiple comparisons test, *$p < 0.001$. **d** The relative amplitude of the corticosterone rhythm (the ratio of the average level at CT 10-18 divided by the average at CT 22-6) in EGFP (black) and ChR2 (blue) mice on each day of collection. Blue line depicts the day of optogenetic stimulation from CT 11-13. Two-way repeated-measures ANOVA with post-hoc Sidak's multiple comparisons test, $p = 0.043$. **e** Corticosterone rhythms over 3 days in EGFP (black, $n = 5$) and ChR2 mice (blue, $n = 5$). Lines and shading depict mean ± SEM. Blue bar, time of optogenetic stimulation (8 Hz, 10 ms, 470 nm, 2 h from CT 23-1). Mixed-effects model, $p = 0.929$. **f** The peak-trough amplitude of the corticosterone rhythm in EGFP (black) and ChR2 (blue) mice on each day of collection. Blue line depicts the day of optogenetic stimulation from CT 23-1. Two-way repeated-measures ANOVA, $p = 0.054$. Source data are provided as a Source Data file.

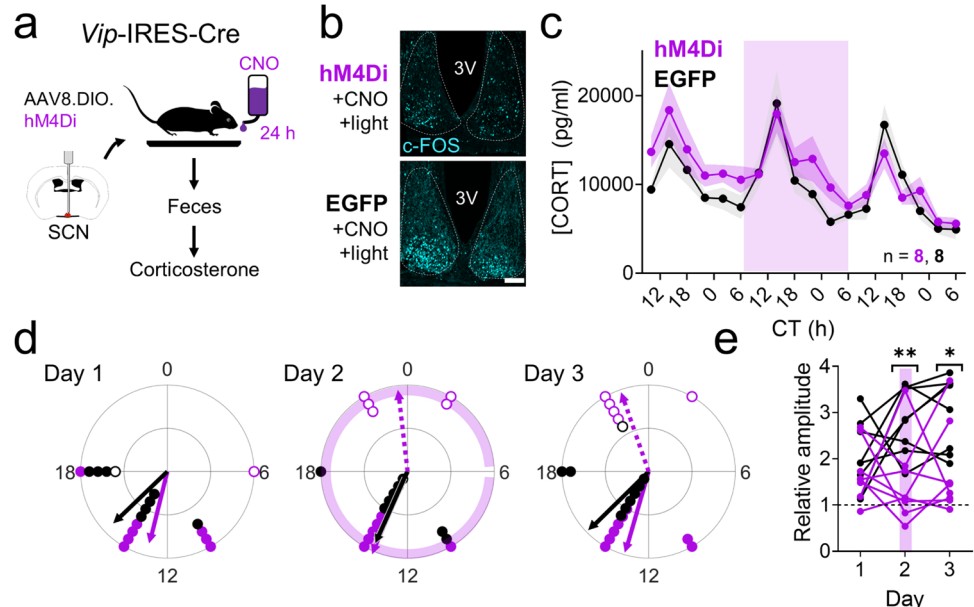

**Fig. 5 Silencing of SCN^VIP neurons augments circadian rhythms in corticosterone during the early morning. a** Schematic for fecal corticosterone collection and chemogenetic SCN inhibition. **b** c-FOS immunoreactivity ($n = 4$ replicates, cyan) after clozapine-N-oxide (CNO) administration 30 min before a 15 min light pulse at CT (circadian time) 12 in $Vip^{Cre/+}$ + hM4Di (hM4Di, top) and $Vip^{Cre/+}$ + EGFP (EGFP, bottom) mice. 3V third ventricle. Scale bar = 100 µm. **c** Corticosterone rhythms over three days in EGFP (black, $n = 8$) and hM4Di mice (purple, $n = 8$). Lines and shading depict mean ± SEM. Purple shading, time of ad libitum exposure to CNO in the drinking water. Mixed-effects model, $p = 0.066$. **d** Rayleigh plots of the first (filled dots, solid lines) and second (open dots, dashed lines) corticosterone peaks in EGFP (black, first peak times days 1–3 CTs 15.0, 13.5, 14.9, Rayleigh test, $p = 0.004, 0.002, <0.001$ on days 1, 2, and 3, respectively; no significant second peak times) and hM4Di (purple, first peak times days 1–3 CTs 12.9, 13.5, 13.1, Rayleigh test, $p = 0.004, <0.001, <0.001$ on days 1, 2, and 3, respectively; second peak times days 2–3 CTs 23.6, 22.7, Rayleigh test, $p = 0.013, 0.007$ on days 2 and 3, respectively) mice on each day of collection. **e** The relative amplitude of the corticosterone rhythm (the ratio of the peak level at CT 10-18 divided by the peak level at CT 22-6) in EGFP (black) and hM4Di (purple) mice on each day of collection. Purple line depicts the day of ad libitum exposure to CNO in the drinking water for 24 h. Two-way repeated-measures ANOVA with post-hoc Sidak's multiple comparisons test, *$p = 0.010$, **$p = 0.006$. Source data are provided as a Source Data file.

consequently, decreased the peak-trough amplitude (Fig. 4d) on the day of stimulation. Increasing SCN^VIP neuron activity at subjective dusk had no effect on the time of the daily peak in corticosterone on the day of or after stimulation (Supplementary Fig. 4b).

Next, we tested whether the effects of SCN^VIP neuronal firing on corticosterone levels depend on time of day. We housed a new cohort of *Vip*-Cre; ChR2 and *Vip*-Cre; EGFP mice for at least 1 week in LD and then collected feces in DD for 3 days of corticosterone measurement every 4 h. On the second day of collection, we optogenetically stimulated the animals around subjective dawn (CT 23-1), before the daily increase in SCN^VIP neuron firing and near the trough of the daily corticosterone rhythm in WT mice. We found that activating SCN^VIP neurons at subjective dawn had no effect on the peak or the peak-trough amplitude of the daily rhythm in fecal corticosterone during or after stimulation (Fig. 4e, f and Supplementary Fig. 4c), but significantly delayed the time of peak corticosterone by ~2 h on the day after stimulation (Supplementary Fig. 4d). Together, these results demonstrate that activating SCN^VIP neurons can acutely suppress corticosterone levels when activated during the evening corticosterone surge.

Because depolarization and hyperpolarization can have opposing effects on shifting circadian rhythms[43,57,58], we next tested the hypothesis that reducing the firing of SCN^VIP neurons would oppose their actions on the phase and amplitude of corticosterone rhythms. We injected the SCN of heterozygous *Vip*-IRES-Cre mice with a Cre-dependent virus encoding the inhibitory chemogenetic construct hM4Di[59] or EGFP as a control. Mice habituated in LD for at least 1 week before being placed into DD

for fecal corticosterone collection every 4 h (Fig. 5a). We first confirmed that we could successfully use chemogenetics to inhibit SCN^VIP neuron activity (Fig. 5b), consistent with prior reports[34]. Next, we inhibited the firing of SCN^VIP neurons by giving mice ad libitum access to the hM4Di ligand clozapine-N-oxide (CNO, 1 mg/kg) in the drinking water for 24 h from CT 6 to CT 6 on the following day (Fig. 5c). Mice of both genotypes consumed between 5 and 8 ml (at least 1 mg/kg) of CNO water with no difference in the amount consumed between genotypes. SCN^VIP inhibition had no effect on the time or amplitude of the daily peak in corticosterone. However, compared to the peak time of their corticosterone rhythm on the day before CNO treatment, 5/ 8 hM4Di mice and 0/8 EGFP mice exhibited elevated corticosterone levels (a "second" corticosterone peak) around CT 22-6, the normal time of the corticosterone trough (Fig. 5d and Supplementary Fig. 5a, b). Consequently, this treatment decreased the peak-trough amplitude in hM4Di compared to EGFP mice on the day of and after inhibition (Fig. 5e). Together, these results demonstrate that chronically reducing SCN^VIP neuron activity increases corticosterone release during the early morning.

**SCN^VIP neurons inhibit PVN^CRH neuron activity and entrain PVN clock gene rhythms.** To determine if these effects of SCN^VIP neuron activity on corticosterone patterns depend on acute or circadian changes in PVN^CRH neurons, we first needed to determine whether SCN^VIP and PVN^CRH neurons are synaptically connected. We performed retrograde and anterograde monosynaptic tracing experiments (Supplementary Fig. 6a–d) but only found sparse synaptic connections between SCN^VIP and

PVN$^{CRH}$ neurons. However, using double-label immunohistochemistry, we found dense VPAC2R labeling throughout the SCN and PVN that overlapped with PVN$^{CRH}$ neurons (Supplementary Fig. 6e), consistent with prior reports[60–63]. These results, in combination with our identification of several SCN$^{VIP}$ axons terminating near PVN$^{CRH}$ neurons (Supplementary Fig. 6f), suggest that SCN$^{VIP}$ neurons volumetrically release VIP to communicate with PVN$^{CRH}$ neurons.

Next, we measured whether SCN$^{VIP}$ neurons directly alter or entrain clock gene expression and calcium activity in PVN$^{CRH}$ neurons. We generated a line of mice that expressed Cre recombinase in *Crh* neurons and Flp recombinase in *Vip* neurons (Fig. 6a). We injected Flp-dependent ChR2 into the SCN and a Cre-dependent, red-shifted genetically encoded calcium indicator, jRCaMP1b[64], into the PVN of individual mice and found SCN$^{VIP}$ neuron terminations near PVN$^{CRH}$ neurons (Fig. 6b). We next imaged calcium in ex vivo PVN slices before, during and after optogenetic stimulation of ChR2-expressing SCN$^{VIP}$ terminals at projected CTs 11-12 (Fig. 6c, d) and found calcium events and integrated calcium levels decreased in the majority of PVN$^{CRH}$ neurons in all PVN slices (Fig. 6e). We then tested the effects of inhibiting SCN$^{VIP}$ neurons activity around midday, their time of peak firing[34], on ex vivo PVN slices. We imaged GCaMP6s-expressing PVN$^{CRH}$ cell bodies while inhibiting hM4Di-expressing SCN$^{VIP}$ terminals with CNO using established in vitro CNO protocols[65]. We found 1 µl CNO (10 µM) pipetted directly onto the PVN slice between CT 6-8 rapidly increased integrated calcium levels, but not calcium events, in the majority of PVN$^{CRH}$ neurons in all PVN slices (Fig. 6f–h). Together, these results demonstrate that SCN$^{VIP}$ neurons can acutely inhibit PVN$^{CRH}$ neurons.

Finally, we made ex vivo explants from triple-transgenic *Vip*$^{Cre/+}$; Ai32$^{fl/+}$; PER2$^{LUC/+}$ mice to record PERIOD2 expression in SCN and PVN cells while stimulating SCN$^{VIP}$ neurons. We found that PER2 circadian rhythms in isolated PVN cultures rapidly damped (Supplementary Fig. 6g), consistent with what we observed for PVN$^{CRH}$ neuron calcium rhythms and what has previously been reported for PVN firing rate and clock gene expression rhythms[44,46]. When we recorded PER2 rhythms from the PVN in slices containing the SCN, we found PVN PER2 rhythms persisted for several days. Daily optogenetic stimulation of SCN$^{VIP}$ neurons for 1 h (470 nm, 8 Hz, 10 ms duration) shifted PER2 rhythms such that after three days of stimulation they were delayed in the SCN and PVN by about 4.5 h compared to controls (Fig. 6i, j and Supplementary Fig. 6h). These results indicate that firing of SCN$^{VIP}$ neurons can entrain the intrinsic daily rhythms of the PVN.

## Discussion

Our results reveal circadian rhythms in gene expression and calcium events in PVN$^{CRH}$ neurons that synchronize to the local light cycle and rely on properly timed signals from SCN$^{VIP}$ neurons to generate the daily surge in corticosterone in anticipation of waking. Consistent with our understanding of a central clock that coordinates peripheral oscillators throughout the brain and body[66], these data redefine the circadian corticosterone circuit to include inhibitory and entraining signals from SCN$^{VIP}$ neurons onto intrinsically circadian PVN$^{CRH}$ neurons. When this timed coordination is lost among networked circadian oscillators in the brain, this results in dysfunctional glucocorticoid release.

We found that, in a light cycle, *Per2* in PVN$^{CRH}$ neurons peaks around midday, followed by a peak in calcium activity a few hours later in the mid-afternoon. These rhythms persist in constant darkness and are dampened in constant light. Thus, the daily rises in *Per2* and calcium follow the reported early morning

peak in *Crh* mRNA[45,67,68] and align with the times of maximal CRH in the hypothalamus (ZT 6-8)[69,70] and median eminence (ZT 8–9)[71,72]. Our long-term in vivo recordings also support prior in vitro results showing higher daytime firing and expression of PER2 protein and *Per1* mRNA in the PVN[46,48,73], and provide the sensitivity, specificity, and spatiotemporal resolution to distinguish between classes of PVN cells. For example, PVN$^{CRH}$ comprise only 15% of all neurons in the PVN[74] and we find they peak within a few hours of each other daily. However *Per2* and *c-fos* mRNA in the PVN have been reported to peak around subjective dusk[45,75], and neurons throughout the PVN have a broad distribution of peak times of activity[43,46,76]. With this first in vivo evidence of a defined population of neurons outside the SCN that function as circadian oscillators according to biologically defined rules of entrainment, we conclude that daily rhythms in calcium and clock gene expression are widespread in the brain and depend on cell type.

Surprisingly, we found that the relative timing of daily *Per2* and calcium activity depends on neuron type. In PVN$^{CRH}$ neurons, *Per2* peaked ~3 h before calcium activity, but in SCN$^{VIP}$ neurons, *Per2* peaks ~6 h after calcium activity[34,50]. Consequently, PVN$^{CRH}$ neurons lag SCN$^{VIP}$ neurons by ~3 h in peak calcium activity and precede them by ~10 h in peak *Per2* expression. In the SCN, many neurons depolarize and fire during the rising phase of *Per2* as a consequence of combined circadian control of transcription of the key leak and voltage-gated potassium channel genes during the night and trafficking of leak sodium channels during the day[3,4,31,77–80]. However, the phase relationship between PER2 and calcium is reversed in SCN astrocytes compared to SCN neurons[81] and, in *Drosophila*, all pacemaker neurons have similar PER rhythms, but calcium peaks at cell-type-specific times of day[82]. Although the cell-intrinsic and network signals that set PVN$^{CRH}$ peak excitability to a few hours after SCN$^{VIP}$ neurons remain to be elucidated, the circadian regulation of excitability appears to differ by cell type.

PVN$^{CRH}$ neurons also differ from SCN neurons in how they synchronize their daily rhythms to local time. Whereas SCN neurons express sustained circadian rhythms and can receive direct retinal input to entrain to ambient light cycles, PVN neurons depend on indirect photic input[83]. Our in vivo and in vitro results show that daily input from SCN$^{VIP}$ neurons to PVN$^{CRH}$ neurons suffices to entrain and sustain the otherwise damped circadian rhythms among PVN neurons. In this regard, rhythms in the PVN may be typical of rhythms in other brain regions that dampen in vitro[44,84]. SCN cocultures have been shown to support daily rhythms in the PVN, perhaps through the synaptic or paracrine release of AVP[46]. However, which signals are integrated by PVN neurons to acutely modulate day-night activity, and which signals act to shift and entrain the PVN clock, are still unknown. Indeed, PVN$^{CRH}$ neurons integrate diverse inputs including daily cues such as light-dark and less-predictable signals such as stressors. For example, we found that PVN$^{CRH}$ neurons increase their activity in response to stress, but SCN$^{VIP}$ neurons do not. SCN$^{VIP}$ neurons greatly increase their activity in response to light at night[34], but PVN$^{CRH}$ neurons do not (Supplementary Fig. 2l). Taken together, these results indicate that fundamentally different circadian oscillators in the PVN and SCN work together to generate and coordinate daily cycles in physiology and behavior.

Daily rhythms in corticosterone release require an intact SCN[7]. Surprisingly, we found that the PVN$^{CRH}$ neuron clock also is necessary. Deletion of BMAL1 in CRH neurons results in low amplitude, irregular corticosterone release. We found that when clockless PVN$^{CRH}$ neurons fail to entrain to SCN signals, their coordinated drive (i.e., calcium activity rhythms) becomes irregular, impeding the daily surge in corticosterone. Our results,

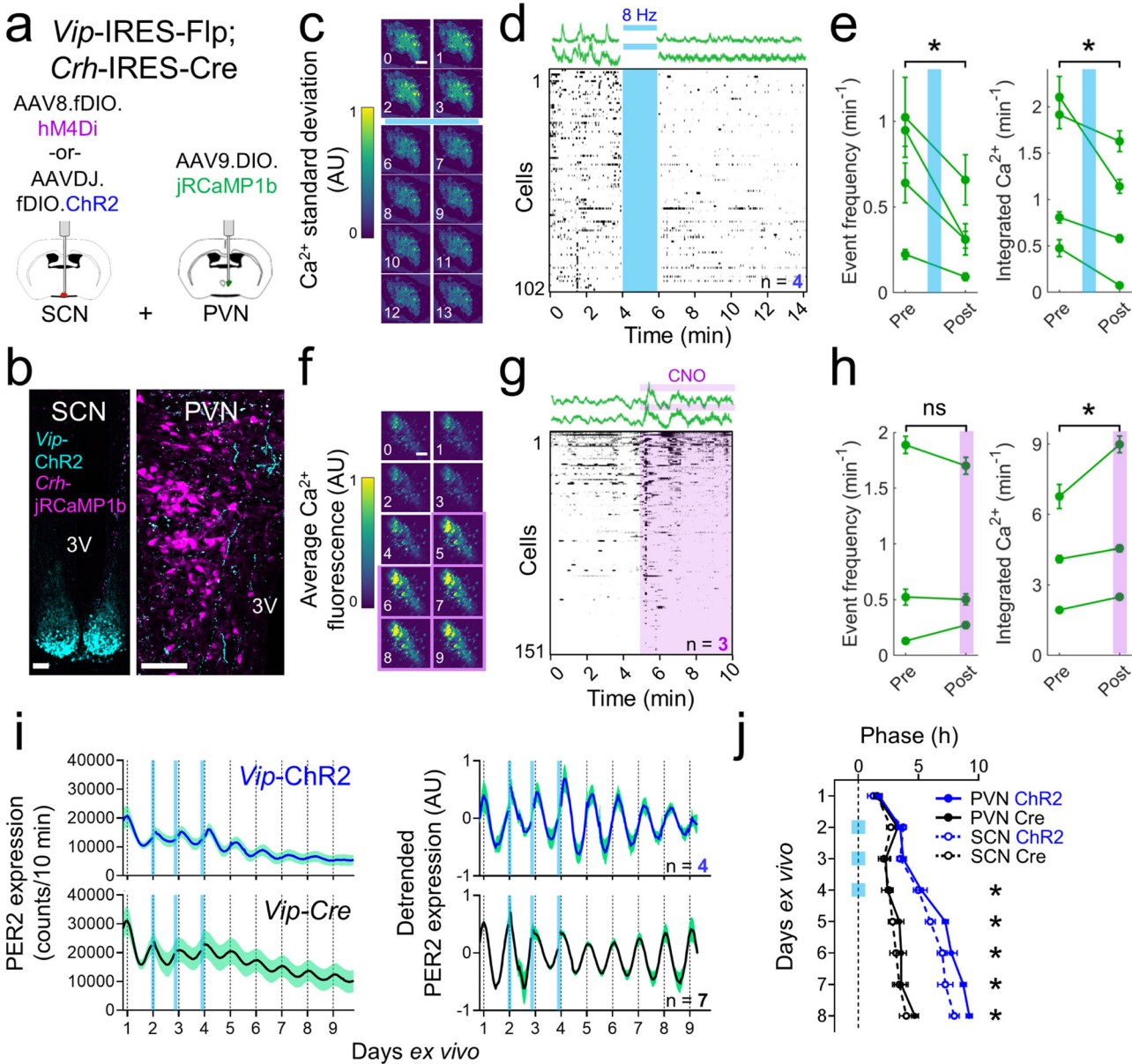

**Fig. 6 SCN$^{VIP}$ neurons inhibit PVN$^{CRH}$ neuron activity and shift PVN clock gene rhythms. a** Schematic for simultaneous SCN$^{VIP}$ neuron manipulation and PVN$^{CRH}$ neuron calcium imaging. **b** Representative ($n = 4$ replicates) images from a $Vip^{Flp/+}$; $Crh^{Cre/+}$ mouse showing concurrent Flp-dependent ChR2 (cyan) and Cre-dependent jRCaMP1b (magenta) expression. 3V third ventricle. Scale bar = 100 μm. **c** The change in calcium fluorescence over each minute of recording in an individual ex vivo PVN slice before and after stimulation (blue bar; 470 nm, 8 Hz, 10 ms pulse width, 2 min duration) around subjective dusk (circadian time (CT) 11-12). Warmer colors represent a greater change in calcium fluorescence. **d** Top, representative calcium traces (in $\Delta F/F$) from two PVN$^{CRH}$ neurons recorded before and after stimulation. Bottom, raster plot depicting PVN$^{CRH}$ neuron calcium activity in individual neurons before and after SCN$^{VIP}$ stimulation ($n = 102$ cells, 4 slices). Raster plot was thresholded to show the upper 50% $\Delta F/F$ of each trace for visualization. **e** Calcium event frequency and integrated calcium levels in PVN$^{CRH}$ neurons in each PVN slice (green lines, mean ± SEM) before and after SCN$^{VIP}$ stimulation (blue line). Two-way repeated-measures ANOVA, $p < 0.001$ for event frequency and integrated calcium levels. **f** The average calcium fluorescence over each minute of recording in an individual ex vivo PVN slice before and during clozapine-N-oxide (CNO) treatment (purple shading, 1 μl of 10 μM CNO) around subjective afternoon (CT 6-8). Warmer colors represent increased calcium fluorescence. **g** Top, representative calcium traces from two PVN$^{CRH}$ neurons recorded before and during CNO treatment. Bottom, raster plot depicting PVN$^{CRH}$ neuron calcium activity in individual neurons before and after CNO treatment ($n = 151$ cells, 3 slices). **h** Calcium event frequency and integrated calcium levels in PVN$^{CRH}$ neurons in each PVN slice (green lines, mean ± SEM) before and during SCN$^{VIP}$ inhibition (purple shading). Two-way repeated-measures ANOVA, $p < 0.001$ for integrated calcium levels, $p = 0.345$ for event frequency. **i** (Left) Raw and (right) detrended PER2::LUC bioluminescence traces from the PVN of $Vip^{Cre/+}$; PER2$^{LUC/+}$ (black, $n = 6$) and $Vip^{Cre/+}$; Ai32$^{fl/+}$; PER2$^{LUC/+}$ (blue, $n = 4$) mice stimulated for 1 h per day for 3 d (blue bars, 8 Hz, 470 nm, 10 ms). **j** Peak times of PER2::LUC bioluminescence in PVN (solid circles and lines) and SCN (open circles, dashed lines) slices from $Vip$-ChR2 (blue) and $Vip$-Cre (black) mice before, during, and after daily optogenetic stimulation (blue squares). Peak times between ChR2 and control PVN were significantly different on days 4-8 (Two-way circular ANOVA, $p < 0.001$ on each day). Source data are provided as a Source Data file.

therefore, place the clock in PVN[CRH] neurons as a point of sensitivity in the circadian corticosterone circuit. Ablating BMAL1 in the entire animal[85] or upstream of PVN[CRH] neurons in the SCN[56] leads to arrhythmic corticosterone release. In contrast, ablating BMAL1 downstream of PVN[CRH] neurons—in the adrenal glands—does not affect corticosterone rhythmicity[86,87] (but see refs. [88,89]). This is surprising as disrupting brain clocks in other neuroendocrine circuits, such as those in kisspeptin or GnRH neurons, has little to no effect on physiology[90,91]. Importantly, the irregular, low daily surge in corticosterone we found in our *Crh*[Cre/Cre]; *Bmal1*[fl/fl] mice is distinct from *Crh* knockout mice, which have chronically low levels of corticosterone[92]. Rhythmic PVN[CRH] neurons are therefore uniquely situated in the HPA axis as a necessary, but not sufficient, component of the circadian corticosterone circuit.

How does loss of BMAL1 in PVN[CRH] neurons disrupt the daily release of corticosterone? Corticosterone rhythms are restored in hypophysectomized rats supplemented with constant ACTH and thyroxine[93,94] and in *Crh* knockout mice infused with constant levels of CRH[92]. This indicates that CRH (and, subsequently, ACTH) is a permissive signal for glucocorticoid release. For example, the coincidence of high levels of ACTH and an autonomic-driven increased sensitivity of the adrenal glands to ACTH[95] in the early evening leads to a peak in corticosterone release before waking. Here, we found that BMAL1-ablated PVN[CRH] neurons lose their daily rhythm in excitability. While this loss of physiological rhythmicity is similar to that seen in SCN neurons lacking clock gene rhythms[96–98], rhythmic SCN drive to BMAL1-ablated PVN[CRH] neurons is maintained in our recordings. This indicates that the SCN requires intrinsic circadian rhythms in PVN[CRH] neurons to drive coordinated rhythms in corticosterone release. We predict that arrhythmic PVN[CRH] neurons release CRH at irregular times which acts on the pituitary and, perhaps, on PVN pre-autonomic neurons that project to the adrenal glands[99], leading to mistimed ACTH release and adrenal sensitivity to ACTH. Our results, therefore, suggest that intrinsic, circadian gene expression in PVN[CRH] neurons that underlie their rhythms in excitability and response to SCN input is required for the appropriately-timed, high-amplitude release of CRH that is fundamental for normal corticosterone rhythmicity.

One caveat to these results is that our transgenic *Crh*[Cre/Cre]; *Bmal1*[fl/fl] mice have BMAL1 ablated in all CRH neurons, not just those in the PVN (Supplementary Fig. 3j). However, because electrolytic lesion or surgical isolation of the PVN that spare all other brain nuclei dramatically reduces circadian corticosterone release[100–102] and conditional knockout of *Crh* in the hippocampus, bed nucleus of the stria terminalis, and central nucleus of the amygdala has no effect on the circadian or stress-induced release of corticosterone[103,104], we predict that BMAL1 ablation in extra-PVN CRH neurons would have little effect on the circadian corticosterone rhythm.

SCN-derived VIP has long been postulated to be the missing excitatory signal to the PVN that stimulates daily corticosterone release[41,42,105,106]. Instead, we found that activating SCN[VIP] neurons around dusk (when SCN[VIP] neuron activity has declined[34,36,43]) blunts corticosterone release and inhibits PVN[CRH] neuron activity. We also found that inhibiting SCN[VIP] neurons in the morning (when SCN[VIP] neuron activity is increasing) or at midday augments corticosterone release and increases PVN[CRH] neuron activity. Conversely, activating SCN[VIP] neurons around dawn (when SCN[VIP] neuron activity is increasing) or inhibiting SCN[VIP] neurons during the early night (when SCN[VIP] neurons are inactive) has no effect on the amplitude of the corticosterone rhythm. Thus, SCN[VIP] neuron activity during both the morning and afternoon suppresses PVN[CRH] neuron activity, the corticosterone rhythm, and, most

likely, CRH release. Intriguingly, blocking SCN-derived AVP release in the PVN during the morning also augments corticosterone release[106]. This suggests that inhibitory SCN[AVP] and SCN[VIP] inputs to PVN[CRH] neurons are each necessary, but not individually sufficient, to suppress corticosterone during the day. Although this model is contrary to prior predictions about VIP, our results are consistent with a recent report showing that inhibiting SCN[VIP] neurons around midday acutely increases corticosterone levels[43].

Importantly, we also found that activating SCN[VIP] neurons can also entrain the clock in PVN neurons. Thus, the action of SCN[VIP] neurons on PVN[CRH] neurons and corticosterone release is both acute and circadian. These actions may be separate (e.g., masking vs. entrainment), may depend on different signaling pathways (e.g., neuropeptides vs. neurotransmitters, synaptic vs. paracrine release), and may be mediated by VIP, GABA, or other signals from SCN[VIP] neurons. Our tracing and immunohistochemistry results suggest that SCN[VIP] neuron axons that terminate in the PVN must volumetrically release VIP onto VPAC2-expressing PVN[CRH] neurons. Indeed, prior studies have shown that *Vipr2* mRNA is expressed in a subset of PVN[CRH] neurons[107] that receive VIPergic projections[108]. PVN[CRH] neurons may also respond to GABA directly or indirectly from SCN[VIP][43] or other SCN neurons[109]. Regardless of the signaling mechanism, our results clearly demonstrate that SCN[VIP] neurons entrain the PVN and corticosterone rhythms through a daily inhibitory signal.

Our results raise the possibility that there is no stimulatory signal from the SCN to the PVN. Instead, we predict that in vivo, SCN[VIP] neuron activity provides a weak inhibitory input throughout the subjective day to suppress PVN[CRH] neuron activity in the morning and evening. Intrinsic, BMAL1-dependent rhythms in PVN[CRH] neuron depolarization around midday may suffice to overcome SCN[VIP] and SCN[AVP][30] inhibition. Alternatively, there may be an unidentified midday excitatory signal from the SCN to the PVN[106]. Overall, this balance of excitation and inhibition entrains rhythms in PVN[CRH] neurons to peak 2–3 h after SCN[VIP] neurons and ~4 h before maximum corticosterone release.

Glucocorticoid release ultimately depends on parallel signaling pathways in the PVN, including monosynaptic projections from SCN[VIP] neurons to pre-autonomic PVN neurons[110] that subsequently project to the adrenal glands. This pre-autonomic pathway, for example, mediates the acute light-induced suppression of corticosterone around dusk[26], similar to the suppressed corticosterone peak that we observed with stimulation of SCN[VIP] neurons. Together, these SCN-to-PVN circuits shape the circadian corticosterone release profile so that it peaks just before waking.

Our results establish a new model for the circadian regulation of corticosterone that includes a daily inhibitory signal from SCN[VIP] neurons to circadian populations of PVN[CRH] neurons. A caveat of our model is that these experiments were performed in male mice. Female mice exhibit fecal corticosterone rhythms that are modulated by estrous state;[111] it is unclear whether the SCN[VIP]-PVN[CRH] neuron regulation of corticosterone differs between sexes. Future studies should also investigate the relative contribution of local clocks in pre-autonomic PVN neurons and in the pituitary to daily corticosterone production. Intriguingly, a recent study found that ablating BMAL1 in all PVN neurons dampens metabolic rhythms and causes obesity[112]. As the PVN[CRH] clock is essential for circadian corticosterone release, and as corticosterone rhythms can synchronize clocks throughout the brain and body[113,114], we predict that rhythms in PVN[CRH] neurons are essential for synchronous circadian rhythms across different organs and, consequently, health.

## Methods

**Animals**. After weaning and prior to recording, we housed male mice in a 12 h:12 h light/dark (LD, where lights on is defined as zeitgeber time (ZT) 0; light intensity ~2 × 10^14 photons/cm^2/s) cycle with food and water provided ad libitum and constant temperature (~22 °C) and humidity (~40%). In all experiments, mice were at least 6 weeks old at the time of virus injection and ~2–5 months old during recordings. For fluorescent and bioluminescent fiber photometry recordings, we used heterozygous *Crh*-IRES-Cre knock-in mice (*Crh*^Cre/+^)[115] on a C57BL/6JN background or homozygous *Crh*-IRES-Cre mice crossed to homozygous *Bmal1*^fl/fl^ mice[54] on a C57BL/6JN background to yield *Crh*^Cre/Cre^; *Bmal1*^fl/fl^ mice. For BMAL1 ablation with corticosterone collection experiments, we used *Crh*^Cre/Cre^; *Bmal1*^fl/fl^ mice. As controls, we used either homozygous *Crh*^Cre/Cre^ (*n* = 3), homozygous *Bmal1*^fl/fl^ (*n* = 4), or heterozygous *Crh*^Cre/+^; *Bmal1*^fl/fl^ mice (*n* = 6). For experiments manipulating SCN^VIP^ neurons and measuring corticosterone, and for in vitro tract-tracing experiments, we used heterozygous *Vip*-IRES-Cre mice[115] on a C57BL/6JN background. For a subset of tracing experiments, we used heterozygous *Vip*-IRES-Cre mice crossed to heterozygous Ai14 (floxed tdTomato) mice[116] on a C57BL/6JN background. For ex vivo experiments, we used *Crh*^Cre/+^ mice crossed to heterozygous *Vip*-IRES-Flp mice[117] backcrossed to a C57BL/6JN background for at least 4 generations to yield *Crh*^Cre/+^; *Vip*^Flp/+^ mice and *Vip*-IRES-Cre mice crossed with floxed Ai32[118] and PER2::LUC[119] mice to yield triple-transgenic *Vip*^Cre/+^; Ai32^fl/+^; PER2^LUC/+^ mice on a C57BL/6JN background as previously published[35]. All primers used for genotyping are listed in Supplementary Table 2. All experiments were approved by and performed in accordance with the guidelines of Washington University's Institutional Animal Care and Use Committee.

**Virus injection and stereotaxic surgery**. We used the following viruses in our experiments: AAV9.CAG.FLEX.GCaMP6s (Penn Vector Core; final concentration of ~1 × 10^13 genome copies (GC)/ml); AAV9.CAG.FLEX.EGFP (Penn Vector Core; ~1 × 10^13 GC/ml); AAV8.DIO.hChR2(H134R).EYFP (Penn Vector Core; ~6 × 10^12 GC/ml); AAV8.hSyn.DIO.hM4D(Gi).mCherry (Addgene, ~2 × 10^12 GC/ml); AAV9.sCAG.DIO.NES.jRCaMP1b (University of Zurich Viral Vector Facility, ~5 × 10^12 GC/ml); AAVDJ.Ef1a.fDIO.hChR2(H134R).EYFP (UNC Vector Core, ~6 × 10^12 GC/ml); AAVrg.Ef1a.DO.DIO.tdTom.EGFP (Addgene, ~8 × 10^12 GC/ml), and AAVDJ.hSyn.FLEX.Synaptophysin-EGFP (University of Zurich Viral Vector Facility, ~6 × 10^12 GC/ml). We also obtained pAAV.*Per2*.DIO.Venus and pAAV.*Per2*.DIO.LUC plasmids as generous gifts from Dr. Eric Erquan Zhang (NBS)[50] and packaged them into Cre-inducible AAV8 viruses (Washington University Viral Core, ~1 × 10^13 GC/ml each). To inject viruses, we placed anesthetized mice (1.5% isoflurane) into a stereotaxic frame on a heating pad to maintain their body temperature throughout the procedure. We then infused virus to the PVN (−0.80 mm posterior, ±0.20 mm lateral, −4.65 mm ventral from bregma) or the SCN (−0.46 mm posterior, ±0.15 mm lateral, −5.65 mm ventral from bregma) at a rate of 0.05 μl/min through a 30-gauge needle attached to a 1 μl syringe (Neuros; Hamilton, Reno, NV). We injected a total volume of 0.3 μl unilaterally (GCaMP6s, *Per2*-Venus, *Per2*-LUC, EGFP) for fiber photometry experiments, 0.25 μl bilaterally (ChR2) for optogenetics experiments, 0.5 μl bilaterally (hM4Di) for chemogenetics experiments, 0.3 μl unilaterally (CreSwitch, Syp-EGFP) for tracing experiments, and 0.3 μl unilaterally (jRCaMP1b) and 0.25 μl bilaterally (ChR2) for ex vivo experiments. After infusion, we left the needle in place for at least 8 min before slowly withdrawing over the course of 1 min. For fiber photometry experiments, we subsequently implanted a fiber optic cannula (5 mm in length, 400 μm diameter core, 0.48 NA; Doric Lenses, Quebec, Canada) immediately dorsal to the virus injection site. We secured the cannulas with a layer of adhesive cement (Parkell, Edgewood, NY) and opaque black dental cement and covered the dental cement with a thin layer of black nail polish. We then transferred mice to their home cages for ~4 weeks to allow for recovery and virus expression.

**In vivo fiber photometry recording**. After recovery and virus expression, we transferred mice to individual open-topped cages maintained in a temperature-, humidity-, and light-controlled chamber (light cycles: 12 h:12 h LD, constant darkness (DD), or constant light (LL); light intensity ~2 × 10^14 photons/cm^2/s) with food and water provided ad libitum. For bioluminescent fiber photometry experiments, we subcutaneously implanted an osmotic pump (Alzet 1004) containing D-luciferin K (100 mM; Goldbio, St. Louis, MO). We chronically tethered freely-moving mice to a fiber optic cable (400 μm core, 0.48 NA, 1.5 m length; Doric Lenses) connected to the implanted optic fiber by a ceramic mating sleeve. We began fiber photometry recording after mice had at least 7 d to acclimate to being tethered using established methods[34]. Briefly, for fluorescent fiber photometry experiments, blue (490 nm) and violet (405 nm) LED lights (Thorlabs, Newton, NJ) were sinusoidally modulated at 211 and 531 Hz, respectively, using a custom Matlab program (Mathworks, Natick, MA) and a multifunction data acquisition device (National Instruments, Austin, TX). The excitation lights passed through a fluorescence cube containing two separate excitation filters (450–490 nm and 405 nm), reflected off a dichroic mirror, and coupled into the tethered fiber optic cable. We used an optical power meter (Thorlabs) to set the light intensity for each wavelength to ~30 μW at the end of the fiber optic cable. Sinusoidally modulated fluorescence was collected through the same fiber optic cable, passed through an emission filter (500–550 nm), and focused onto a photoreceiver

(Newport, Irvine, CA). The photoreceiver signal was then sent to two lock-in amplifiers (Stanford Research Systems, Sunnyvale, CA) that were synchronized to 211 or 531 Hz for demodulation. We collected voltage signals from the amplifiers at 10 kHz using a custom Matlab program and a multifunction data acquisition device and saved them to disk as binary files. For calcium activity recordings, we automatically recorded GCaMP6s fluorescence for 10 min per h for multiple days. For clock gene expression recordings, we automatically recorded Venus fluorescence for 60 s per 15 min for multiple days. We excluded animals from analysis if the 490 and 405 nm signals simultaneously fluctuated by ~100 mV (typically indicating fiber placement inside the third ventricle), if they did not exhibit circadian locomotor activity, or if the fiber or virus was off-target as determined by histology. For bioluminescent fiber photometry experiments, tethered animals housed in constant darkness were connected to a custom-built photomultiplier tube-photon counting head assembly (Hamamatsu H9319). Bioluminescence counts were measured over 10 min and recorded to disk using custom software.

**Fiber photometry data analysis**. Data were analyzed using established methods[34]. For calcium activity analysis, we fit the calcium-independent isosbestic signal (violet) to the calcium-dependent signal (blue using a linear least-squares fit). We then calculated change in fluorescence over baseline fluorescence (ΔF/F) as

$$\frac{\text{Blue signal} - \text{fitted violet signal}}{\text{Fitted violet signal}}$$

and adjusted the final values so that a ΔF of 0 corresponded to the 2nd percentile of the signal. We converted ΔF/F values to events by using a peak-finding algorithm (Matlab) to count instances longer than 1 s when ΔF/F exceeded the median plus two standard deviations of the trace or 1.5% ΔF/F, whichever was greater. We also calculated the integrated calcium value of each trace by integrating the ΔF/F values above 0% over 10 min. For clock gene expression analysis, because *Per2*-Venus does not have an isosbestic point, we excluded the violet signal from analysis (and used it as a negative control, Supplementary Fig. 1c). We averaged the voltage signal from the blue channel over 60 s for each recording to obtain a single voltage value at each timepoint, fit the data with an exponential filter, and subtracted the fitted signal from the raw data to obtain Venus fluorescence values over time in arbitrary units.

**Ex vivo PVN^CRH^ calcium recording**. We obtained PVN slices using methods modified from ref. [44]. We decapitated anesthetized mice and cut their brains into 250 μm coronal slices using a vibrotome. We then dissected and cultured bilateral PVN individually on cell culture membranes (Millicell-CM, Millipore, Burlington, MA) in a recording medium (DMEM supplemented with 10 mM HEPES, 100 U/ml penicillin/streptomycin, 4.5 g/l D-glucose, and 2% B27) placed in sealed Petri dishes. We placed dishes on an inverted Tsscope (TE2000-S; Nikon, Tokyo, Japan) within an incubation chamber held at 35°C. We then automatically imaged GCaMP6s fluorescence in PVN^CRH^ neurons using a digital EMCCD camera (iXon DU-897; Andor Technology, Belfast, UK) once per hour for 48 h using a custom Micro-Manager script. We measured the fluorescence values of individual static, cell-sized ROIs using ImageJ and a custom Python script, detrended the traces by division, and presented the data as a raster plot.

**Fecal pellet collection and corticosterone analysis**. We placed mice in custom-built collectors maintained in a temperature-, humidity-, and light-controlled chamber with food and water provided ad libitum to allow for non-invasive measurement of fecal corticosterone metabolites from individual mice, a validated method for measuring circadian rhythms in corticosterone release[88,120] These collectors consist of a 20 cm × 20 cm plexiglass cage with a wire bottom placed over a funnel coated in silicone lubricant (#300012, WD-40, San Diego, CA) set above a slotted dish that automatically rotated to allow fecal pellets to fall into a new slot positioned under the funnel every hour. We monitored locomotor activity in these cages using a beam-break sensor positioned in front of the food hopper connected to Clocklab software (v6.1, Actimetrics, Evanston, IL). We allowed mice to acclimate to the cages in LD for at least 7 d before beginning fecal pellet collection, after which they were placed into DD for 24 h. Every 24 h after the start of collection, we removed the collected fecal pellets from the dish, placed them into individual tubes, and baked them at 62 °C until completely dry, typically 2–3 days. We calculated the total weight per timepoint, and we estimate that our collector cages allowed us to recover ~85–100% of total feces produced per hour. To account for any timepoints with no fecal pellets, we binned timepoints into four-hour groups for further processing and ground the resultant samples into a fine powder using a mortar and pestle. We weighed out 20 mg of each sample, added 1 ml of 80% methanol, and agitated for 40 min to extract steroids. After centrifuging, we transferred the supernatant to new tubes and placed them in a fume hood until the methanol fully evaporated, typically 6–7 days. We resuspended the dried samples in 200 μl enzyme-linked immunosorbent assay (ELISA) buffer (Cayman Chemical, Ann Arbor, MI) and subsequently diluted the resuspended samples 1:50 (determined empirically based on the standard curve) in ELISA buffer. We processed the samples in duplicate for corticosterone concentration using the instructions in the ELISA kit (Corticosterone ELISA Kit, Cayman Chemical). We subsequently measured absorbance values using a microplate reader at 415 nm (iMark; BioRad,

Hercules, CA) and determined the final corticosterone concentrations (in pg/ml) based on the standard curve and dilution factor. There were no measurable differences in fecal weight between groups across all experiments. For our $Crh^{Cre}$; $Bmal1^{fl/fl}$ and optogenetics phase plots, peak times of corticosterone were defined as the time of maximum corticosterone on each day of collection. When two timepoints on a given day differed in their corticosterone levels by <15%, we calculated the corticosterone peak as the midpoint between those two timepoints. Because of the multiple peaks in our chemogenetics data, corticosterone peaks in Fig. 5 and Supplementary Fig. 5 were detected using a peak-finding algorithm in Matlab with identical parameters for each animal.

**In vivo activation of SCN$^{VIP}$ neurons.** We transferred control and ChR2-expressing mice to our custom-built feces collectors and chronically tethered them to a fiber optic cable (400 μm core, 0.39 NA, 1.5 m length; Thorlabs) connected to a high-powered 470 nm LED (Thorlabs) under the control of an LED driver. We used an optical power meter to set the light intensity at the fiber tip to ~10.8 mW when driven at 1000 mA. We allowed mice to acclimate to the cages and tethering in LD for at least 7 d before beginning fecal pellet collection. We automatically collected fecal pellets over three days every hour. On the second day of collection, we optogenetically stimulated mice (470 nm, 8 Hz, 10 ms duration) from CT 11-13 or 23-0 using a stimulus generator (Grass Technologies, West Warwick, RI) attached to the LED driver under the control of a light timer.

**In vivo inhibition of SCN$^{VIP}$ neurons.** We transferred control and hM4Di-expressing mice to our custom-built feces collectors and allowed them to acclimate to the cages in LD for at least 7 d before beginning fecal pellet collection. We replaced the water bottles on the cages with 1% sucrose water daily for 3 d prior to the start of collection to acclimate mice to the removal and replacement of their water bottles. We then automatically collected fecal pellets for three days. From CT 6 on the second day of collection until CT 6 the following day, we replaced the water bottles on all cages with clozapine-N-oxide (CNO, 1 mg/kg; Hello Bio, Princeton, NJ) in 1% sucrose water (to mask the bitter taste of CNO). This method of continual CNO administration in the drinking water has been previously shown to successfully alter neuronal activity for the duration of CNO administration[121,122]. Virus expression was similar between mice. Mice consumed between 5 and 8 ml of CNO water, receiving at least 1 mg/kg CNO throughout the day; there was no genotype difference in the amount consumed.

**Ex vivo SCN$^{VIP}$ manipulation and PVN$^{CRH}$ calcium imaging.** We obtained and cultured PVN slices as described above, after which we kept them at 35 °C for at least 1 h before imaging at projected CTs 11-12 or CTs 6-8. We imaged jRCaMP1b or GCaMP6s fluorescence for up to 15 min total using a TRITC or EGFP filter cube and a digital CCD camera (QIClick, QImaging, Tuscon, AZ) at 5 frames/s with QCapture (QImaging) and CamStudio software. For SCN$^{VIP}$ stimulation experiments, we turned off the imaging excitation light after 4 min and optogenetically stimulated SCN$^{VIP}$ terminals for 2 min (470 nm, 8 Hz, 10 ms duration, ~10.8 mW at fiber tip) with a fiber optic cable (400 μm core, 0.39 NA, 1.5 m length; Thorlabs) connected to a high-powered LED under the control of an LED driver (Thorlabs) positioned directly above the slice. We then stopped stimulation, turned the imaging excitation light back on, and continued imaging for an additional 8 min. For SCN$^{VIP}$ inhibition experiments, we imaged for 5 min, pipetted 1 μl CNO (10 μM) directly onto the PVN slice, and continued imaging for an additional 5 min. We verified presence of EYFP (ChR2) or mCherry (hM4Di) expression in SCN$^{VIP}$ terminals in PVN slices and SCN$^{VIP}$ cell bodies in adjacent SCN slices at the end of each experiment. We measured the fluorescence values of individual static, cell-sized ROIs using ImageJ, detrended the traces by division, and calculated change in fluorescence over baseline fluorescence ($\Delta F/F$) as

$$\frac{F_i - \text{median}(F)}{\text{median}(F)}$$

We then converted $\Delta F/F$ values to events by using a peak-finding algorithm (Matlab) and calculated the integrated calcium value of each trace as described above. We excluded from analysis a subset of slices that had good jRCaMP1b expression (healthy cells, no nuclear filling) but exhibited no calcium dynamics before or after stimulation ($\Delta F/F_{pre}$ and $\Delta F/F_{post} = 0 \pm 0.01$).

**Ex vivo clock gene bioluminescence imaging.** We recorded bioluminescence from SCN-PVN cocultures using established methods[123]. We obtained and cultured slices containing the SCN and PVN as described above and transferred them to a light-tight incubator at 36 °C (Onyx, Stanford Photonics, Palo Alto, CA). We collected images with an electron-multiplying CCD camera (iXon, Andor Technology) with an integration time of 1 h. After ~3 d of imaging, we briefly paused our recordings, optogenetically stimulated the cocultures using a custom-built LED array (470 nm, 10 ms, 8 Hz, 1 h), and resumed imaging. We repeated this stimulation paradigm every 24 h for 3 d and then continued imaging for several additional days. For analysis, we loaded the movies in ImageJ, discarded the first 24 h of recording and the frames immediately before and after the time of stimulation, and drew individual ROIs around each PVN and SCN. We applied adjacent frame minimization to each movie to filter out camera noise or cosmic rays. We

detrended raw pixel intensity values using a 4 h moving average, and detected peak times independently using a peak-finding algorithm (Matlab) and crossover analysis[124]. To identify phase shifts, peak times on each day were compared using circular statistics.

**Retrograde and anterograde tracing.** For retrograde monosynaptic tracing experiments, we unilaterally injected the PVN of $Vip^{Cre/+}$ with 0.3 μl AAVrg.E-f1a.DO.DIO.tdTom.EGFP (CreSwitch) virus[125] in the absence of Cre and EGFP in the presence of Cre. After transfection, starter neurons in the PVN and Cre⁻ projection neurons in the SCN would express tdTom, and $Vip$-Cre⁺ projection neurons in the SCN would express EGFP. For anterograde tracing experiments, we unilaterally injected the SCN of Vip$^{Cre/+}$; Ai14 mice with 0.3 μl AAVDJ.hSyn.FLEX.Synaptophysin-EGFP[126] that expresses EGFP-fused synaptophysin in Cre⁺ neurons. After transfection, $Vip$-Cre⁺ axons expressing tdTomato and synaptic terminals expressing EGFP would label monosynaptic connections between the SCN and PVN. For each experiment, after 4 weeks for virus expression and recovery, we injected colchicine into the lateral ventricles (1 μl per side, 1 μg/μl) of anesthetized mice, allowed the mice to recover, and perfused them 24 h later at CT 6. Brains were processed, and PVN slices were immunostained for CRH, tdTomato, and EGFP and imaged as described below.

**Histology and image analysis.** To verify successful optogenetic activation of SCN$^{VIP}$ neurons, at CT 12, we optogenetically stimulated mice for 1 h, and at CT 14, we anesthetized and transcardially perfused the mice with phosphate-buffered saline (PBS) followed by 4% paraformaldehyde (PFA) as in[35]. To verify chemogenetic inhibition of SCN$^{VIP}$ neurons, we injected mice with 1 mg/kg CNO at CT 11.5, presented mice with a 15 min light pulse at CT 12, and perfused 1 h after the start of the light pulse as in[34]. To verify BMAL1 ablation in PVN$^{CRH}$ neurons, for tract-tracing experiments, and for VPAC2 immunohistochemistry, we injected colchicine into the lateral ventricles (1 μl per side, 1 μg/μl) of anesthetized mice, allowed the mice to recover, and perfused them 24 h later at CT 6. After perfusion, we extracted the brains, post-fixed them overnight in 4% PFA, and cryoprotected them in 30% sucrose. We then obtained 40 μm frozen coronal sections using a cryostat (Leica, Wetzlar, Germany) and performed immunostaining using established methods[34]. We used the primary antibodies goat anti-c-FOS (sc52-G, 1:1000; Santa Cruz Biotechnology, Dallas, TX), guinea pig anti-BMAL1 (AB2204, 1:2000; Sigma Aldrich, St. Louis, MO), rabbit anti-CRH (ab8901, 1:1000; Abcam, Cambridge, UK), guinea pig anti-CRH (T-5007, 1:1000; Peninsula Laboratories, San Carlos, CA), chicken anti-GFP (GFP-1020, 1:1000; Aves Labs, Davis, CA), rabbit anti-mCherry (ab167453, 1:1000; Abcam), and rabbit anti-VPAC2 (ab28624, 1:1000, Abcam) and the secondary antibodies DyLight 488 donkey anti-goat (SA5-10086, 1:500; Thermo-Fisher Scientific, Waltham, MA), DyLight 594 donkey anti-goat (SA5-10088, 1:500; Thermo-Fisher Scientific), Alexa 594 donkey anti-guinea pig (706-585-148, 1:500; Jackson ImmunoResearch, West Grove, PA), Alexa 488 donkey anti-rabbit (711-545-152, 1:500, Jackson ImmunoResearch), Alexa 555 goat anti-rabbit (ab150078, 1:500; Abcam), Alexa 488 donkey anti-chicken (703-545-155, 1:500; Jackson ImmunoResearch), and Alexa 647 goat anti-guinea pig (ab150187, 1:500; Abcam). Brains were processed and stained identically (same antibody concentrations, durations, etc.). After immunostaining, we imaged sections on a confocal microscope (Nikon A1, Tokyo, Japan) as a single z-stack using identical laser and capture settings for each image that was to be quantitively compared. We adjusted brightness and contrast settings identically for sections depicted in representative images.

**Statistics.** We chose sample sizes to be sufficient for statistical analysis based on similar techniques used in previous publications[34,35]. We had no specific methods to blind investigators to genotype or condition during the experiments themselves, but investigators were blinded to the genotype or condition when processing data (e.g, corticosterone samples). We performed data analysis blind to genotype or condition. Mice were randomly assigned to experimental groups when possible (e.g., EGFP vs. ChR2). All statistical tests are presented in Supplementary Table 1. We performed the following statistical analyses in Prism 8.0 (GraphPad, San Diego, CA): Brown–Forsythe ANOVA, two-way repeated-measures ANOVA (or a restricted maximum likelihood mixed-effects model), nested one-way ANOVA, and unpaired two-tailed Welch's t-test. We performed the following statistical analyses using the Circular Statistics Toolbox[127] in Matlab: Rayleigh test, two-way circular ANOVA, and Watson–Williams test. We determined circadian rhythmicity using JTK Cycle[128] in R. We used Shapiro–Wilk and Brown–Forsythe tests to test for normality and equal variances, defined α as 0.05, and presented all data as mean ± SEM.

**Reporting summary.** Further information on research design is available in the Nature Research Reporting Summary linked to this article.

## Data availability

All data generated in this study that support our findings are presented within this paper, its Supplementary Materials, or in the source data. All additional information will be made available upon reasonable request to the corresponding author. Source data are provided with this paper.

## Code availability

Matlab code used in this study has been previously published[34] and is available from the corresponding author upon reasonable request.

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

## Acknowledgements

We thank A. Smith for assistance with corticosterone sample processing and the members of the Herzog lab for discussion and comments on the manuscript. This work was supported by National Institutes of Health Grants R01GM131403 (E.D.H.) and F32 HL133772 (J.R.J.).

## Author contributions

J.R.J. and E.D.H. designed the study. J.R.J., S.C., and D.G.-F. performed the experiments. J.R.J., S.C., and D.G.-F. analyzed the data. J.R.J. and E.D.H. wrote the paper with input from all authors.

## Competing interests

The authors declare no competing interests.
