## [Peer Review File · Nature Communications]

Reviewer comments, initial:

-

Reviewer #1 (Remarks to the Author):

In general, levels of hormones exhibit a robust circadian rhythm until the control of the suprachiasmatic nucleus (SCN). How these rhythms are regulated is not well understood although perhaps the best studies includes the SCN regulation of the HPA axis through the neuron in the paraventricular nucleus (PVN). Still our understanding of this regulatory pathway is rudimentary and has not been examined using modern tools of Neuroscience.

In an LD cycle, the PVN CRH neurons peak in Period2 clock gene expression around midday and peak in calcium activity as measured by GCaMP6 a few hours later. These rhythms persist in constant darkness and dampen in constant light.

The loss of BMAL1 in CRH neurons dramatically reduces the amplitude and precision of the daily rhythm in corticosterone release. Somewhat surprizing, the authors found that the loss of BMAL in the PVN CRH neurons lengthened the free running period of the mice in DD.

They then went on to manipulate the activity of SCN VIP neurons with optogenetic construct ChR2 or inhibit these neurons with inhibitory chemogenetic construct hM4Di. They demonstrated that temporally discrete activation or inactivation of SCN VIP neurons alters the peak amplitude and timing of corticosterone release and PVN CRH calcium activity. In addition, daily SCN VIP activation entrains PVN clock gene rhythms by inhibiting PVN CRH neurons.

Together, the results presented in this ms demonstrate that the amplitude and phase of the daily corticosterone surge depends on coordinated clock gene expression and neuronal activity in both SCN VIP and PVN CRH neurons.

In this work, the authors clarified a critical circuit from SCN VIP-producing neurons to PVN CRH. They carried out this work for the most part using state-of-the-art in vivo imaging techniques in freely behaving mice. As needed, they supplemented this in vivo analysis with work in brain slices. The writing is clear and scholarly. The figures clearly support the main concepts. This circuit is biologically important and dysfunction is associated with a number of diseases. This is just a wonderful study!

We just have a few minor concerns.

Should Fig. 1C be labelled ZT/CT? While we appreciate that the authors are measuring Venus fluorescence (yellow), will yellow be hard to see in the final figure? The same symbols should be used in Fig1F and G.

Another minor point is that the authors use yellow to indicate yellow fluorescence in Fig. 1 but also use yellow bars to indicate day. Maybe just use white bars for day?

In Fig 4C, the effect that we see is a suppression of amplitude not the phase shift.

In Fig. 4D, the blue line looks to be in the wrong place. Also, we are not sure of the format of 4D, it seems to be a messy way of showing the phase shifts so clearly illustrated in the polar plot.

Could that authors add the peak/trough plot to Fig. 4?

Again in Fig4F, we have a hard time seeing the phase shift. The polar plots are clear.

In Fig5, it appears that the increase in amplitude and the shift in phase is largely driven by 2-3 mice. Can the authors provide some explanation? Did these mice drink more water and get a higher dose of CNO?

Reviewer #2 (Remarks to the Author):

Summary

In the current manuscript by Jones et al, the authors describe circadian transcriptional and metabolic rhythms of corticotrophin-releasing hormone (CRH) neurons in the hypothalamic paraventricular nucleus (PVN) in vivo. They then use a variety of opto- and chemogenetic manipulations coupled with behavioral, physiological and imaging assays to determine that these rhythms are driven (at least in part) by output from the vasoactive intestinal polypeptide (VIP) neurons in the suprachiasmatic nucleus (SCN). Through these experiments, they determine that optogenetic activation of SCN-VIP neurons at subjective dusk, but not subjective dawn, suppresses the peak in corticosterone levels. In contrast, chemogenetic inhibition of SCN-VIP neurons increases the relative peak of the rhythm in corticosterone. In both cases, manipulation of the SCN-VIP neurons appears to shift the phase of corticosterone rhythms. This is further investigated by a very elegant two-recombinase system, showing that the two SCN-VIP neuron manipulations bidirectionally alter the activity of PVN-CRH neurons with activation of SCN-VIP neurons inhibiting PVN-CRH neuron activity and inhibition of SCN-VIP neurons leading to an increase in the activity of PVN-CRH neurons. Finally, the authors show that daily activation of the SCN-VIP neurons in an ex vivo slice can entrain circadian rhythms in the PVN.

This work builds on and validates a number of previous publications (particularly Mazuski et al, 2020 and Paul et al, 2020) describing the circadian activity and critical output of SCN VIP neurons to control wider aspects of physiology, particularly control of corticosterone release. The observations that SCN-VIP neuron activation and inhibition bidirectionally alter the activity of neurons in the PVN and that chemogenetic inhibition of SCN-VIP neurons enhances corticosterone production closely mirror and validate the findings of Paul et al, 2020. What makes this manuscript novel is the in vivo circadian characterization of the specific PVN-CRH neuron population and the compelling observation that the phase relationship between the core clock and calcium/neuronal excitability varies with cell-type detailed across Figures 1 and 2. Beyond that limited observation, however, I don't find that the rest of the manuscript presents compelling evidence for the conclusions drawn or really elevates the model beyond what has been published before, particularly in Paul et al, 2020.

The results presented here will be of interest to the circadian field, and more broadly to endocrine researchers. However, in the case of the latter, some of the terminology may need to be made more accessible for a wider audience with additional clarity to the figure legends to aid interpretation. Unfortunately, I do have major concerns over the interpretation of some of the data which is detailed below. In my opinion, I don't think that the conclusions as presented by the authors are supported by the data.

Major Comments

In Figure 2h-j the authors show ex vivo data of calcium recordings from semi-acute PVN slices, claiming a phase coherence between cells in the slice, but can only record rhythms for approximately 36h due to the rapid damping of oscillations. In the first 24 hours of a recording like this, there will be cellular synchronization by exogenous experimental factors. The analysis of phase coherence should be conducted once these effects have had time to dissipate to be able to comment on this aspect of network time keeping. In the current recording configuration, obviously this is not possible due to the rate of PVN damping. An experiment like this would be more convincing had it been done on the SCN co-cultures used in Figure 6 which persist for up to a week.

In the experiment where BMAL1 was deleted in the CRH cells of the PVN (Figure 3), was there any loss of BMAL1 in cells elsewhere in the brain aside from the PVN? In the text, it seems to me, that this is presented as a PVN specific deletion. However, CRH is expressed broadly across the brain as

revealed by intersectional genetic imaging using the same CRH-Cre mouse line (Peng et al, 2017). Reference to this fact should be made clearer in the text as it will affect the interpretation of this data. The authors should also present the extent to which BMAL1 is deleted in other brain areas, and comment on how this might affect the interpretation of the corticosterone and locomotor activity observations.

Is genetic ablation of BMAL1 just blunting CRH production resulting in the loss of corticosterone amplitude observed? Have the authors looked at the levels of CRH (at the mRNA and protein levels) in their knockout model? It may be that killing the clock in these cells suppresses the global production of CRH. A more sensitive intervention to show the necessity of the clock in this circuit would be to manipulate the clock without killing it. Deletion of CK1 ϵ TAU which has been used by this lab previously (Tso et al, 2017) would be a better mechanism to see if mistimed TTFL in the CRH neurons can be overridden by the SCN.

Did the authors test the effect of BMAL1 deletion in the CRH neurons on corticosterone rhythms under an LD cycle. Are these rhythms still disrupted if the SCN cells are receiving a daily retinal input, or is the disruption observed a solely free-running phenomenon? In the discussion, the authors allude to the fact that the relative "weakness" of the BMAL1-deficient condition could be due to an entrainment vs masking mechanism. Alternatively, have the authors tested optogenetic VIP-cellular activation in a CRH-BMAL1 deficient mouse using their Flp/Cre line to try and discriminate between these entrainment and masking phenotypes? This further level of analysis would elevate these findings.

In Figure 3, the authors observe a 1 hour advance in the corticosterone peak in wild-type mice on successive days, but did not observe the same 1 hour advance in wild-type mice in other experiments presented in Figure 4e and h, and in Figure 5e. Is this something specific to the mixed wild-type cohort used here (in which case, they should be separated) or is it due to some inherent inaccuracy in the sampling method?

In the optogenetic experiments presented in Figure 4, the authors claim to observe phase shifts in the peak of corticosterone in the 3rd day following stimulation of the SCN-VIP cells at CT11-13. I struggle to see this in the traces presented (Figure 4c) - on the 3rd day, the peak of the mean aggregate trace of ChR2 animals is aligned with the mean aggregate trace of EGFP animals. This largely chimes, by eye, with the fits to the data presented in Supplementary Figure 4c. The same also follows for my interpretation of the raw and aggregate data presented for the optogenetic stimulation at CT23-1 (Figure 4f and Supplementary Figure 4g) and the chemogenetics (Figure 5c and Supplementary 5c). In the methods, it is stated that the corticosterone oscillations are fit using a cosinor approach. If the phases are taken from these fits they will invariably have an error term associated with them which will affect the precision of the determined peak. It would be better if the authors determined the peak in the corticosterone from the raw data rather than from these fits which will have some associated inherent error. Due to the inconsistency in what is presented and what is interpreted, I have trouble seeing where the reported phase shifts come from?

Additionally, the authors report phase shifts of between 90 minutes and 2 hours: how have they come to these numbers? The phases are calculated on a single cycle based on data where the sampling rate, and therefore resolution, is 4 hours. Phase shifts of these magnitudes seem non-sensical.

For the CT11-13 optogenetic data, the authors state in the discussion that the locomotor activity rhythms and corticosterone rhythms dissociate based on locomotor activity presented in a previous paper (Mazuski et al, 2018) and the corticosterone data presented here. However, in Supplementary Figure 4d the authors present locomotor activity rhythms (actograms) for the same mice that were sampled for the corticosterone monitoring experiment: if it is the case that these two rhythms dissociate why not use this data to present phase differences on an animal-by-animal basis? The authors should not be drawing conclusions based on different experimental cohorts, especially when they should have the relevant data to hand, presented within a Supplementary Figure in the manuscript. This data is presented for the CT23-1 optogenetics and chemogenetics (Supplementary Figures 4j and 5f respectively), but not for this critical time point

where the authors speculate on this dissociation. Inclusion and analysis of this data would strengthen this interpretation and provide a new, novel aspect to this work.

Further to this previous point, from the optogenetics at CT23-1 the authors state that there was no effect of stimulation to shift the onset of locomotor activity, but that the corticosterone peak was delayed by approximately 2 hours. They then state that there is no dissociation between the activity rhythms and corticosterone at this time point as the correlation between the two remains the same. Surely, if the peak of corticosterone has moved independent of the activity onset, then there must have been a dissociation between the two?

In the ex vivo work presented in Figure 6i-j, the authors state that recordings of PER2::LUC bioluminescence in the SCN were made in parallel to the recordings made in the PVN. This data is not presented. In my opinion, it would strengthen the analysis of the findings if the authors were able to determine any relative phase or period difference between the SCN and the PVN, as well as commenting on any potential lags in phase shifting between the SCN and the PVN - does the PVN immediately follow the SCN phase, or is there a lag between the two?

In the entrainment conditions it looks, by eye, in both the example trace (i) and grouped data (j) that the Chr2 condition has a shorter period than the non-Chr2 controls that is accounting for the relative difference in entrainment. This is evident in the fact that control lines appear to be diverging from the Chr2 lines from the 2nd day, before Chr2 entrainment begins. Have the authors analyzed or corrected the data for potential differences in the periods of these cohorts.

Minor Comments

In the data presented in Figure 1, did the authors record behavior simultaneous to the fiber photometry for PVN-CRH neurons? If so, out of interest, how did the period of locomotor activity relate to the period of the PVN-CRH neurons, what was the relative phasing between the two rhythms and did this phase relationship change with different lighting conditions?

Line 291: "isolated PVN cultures rapidly damped (n=5, data not shown)". Why not show this data in the Supplementary Figures? This would strengthen the case that the SCN is entraining rhythmicity within the PVN in the SCN-PVN slices.

Line 360-361: "SCNVIP neurons greatly increase their activity in response to light at night (Jones et al, 2018), but PVNCRH neurons do not (n=3, data not shown)." What is the rationale for not showing this data? The data previously presented (here and in Paul et al, 2020) predicts that increased activity in the SCN-VIP neurons will acutely decrease activity in the PVN-CRH neurons, so why is it surprising that the PVN-CRH neurons do not increase as well? Has this data been examined for a decrease in event frequency as the model would predict?

In the discussion, the section relating to the effects of PVN-CRH BMAL1 ablation (lines 367-395) neglects to discuss the effects of BMAL1 ablation outside of the PVN in other CRH neurons and whether BMAL1 ablation may affect the expression of CRH.

Relating to supplementary Figure 3: The labeling of the colors is difficult to discern in the immunohistochemistry (a). The range of the axes cuts off the peak in some of the WT data (c). The statistics occlude the data points (e). There is no description of the dashed red lines in the corticosterone fits (f). In the legend, μM is used in place of μm .

Relating to supplementary Figure 4: The range of the axes cut off the peak in some of the EGFP corticosterone data (b). There is no description of the dashed red lines in the corticosterone fits (c). The significance bracket is offset from the data (j).

Data and Methodology

On the whole, the methodology is well described, but there are some gaps:

In the methods, I could not find reference to the breeding strategy or fiber luminometry methodology used to generate the in vivo bioluminescence data presented in Supplementary Figure 1c.

Details of the experimental approach (including the AAV vector) to perform the chemogenetic inhibition experiments in Figure 6 are not present. What AAV vectors were used? What was the genetic background of the animals from which slices were made?

Appropriate use of statistics and treatment of uncertainties

The legends to the majority of figures do not contain sufficient detail and are not clear enough to allow a non-expert in the circadian field to interpret the figures properly. There is a lot of description and interpretation of the data within the legends – this should be left to the main text. The legends are missing vital information to aid with the reader's own interpretation in isolation e.g. the definition of error bars and other figure components etc - a stated requirement of manuscript format.

In Figure 3, the wild-type cohort are made of a mix of genetic backgrounds. The specific breakdown of the strains/numbers contributing should be listed.

Largely the statistical tests are appropriate to the data used. However, the tests presented in Figure 6e and 6h are, in my opinion, inappropriate. In these tests, the individual cells are treated as independent biological replicates - this is incorrect. These individual cells came from 3 biologically independent mice: the test should reflect this fact. A better approach would be to use a model where the relative contributions of individual cells can be taken into consideration, but where they are not treated as biological replicates for example a linear mixed model approach.

In Supplementary Table 1, the statistical results for some of factors are missing from the table.

References

- Peng, J. et al, 2017. *Front Neuroanat*, <https://doi.org/10.3389/fnana.2017.00063>
Mazuski, C. et al, 2020. *J Biol Rhythms*, <https://doi.org/10.1177/0748730420932073>
Mazuski, C. et al, 2018. *Neuron*, <https://doi.org/10.1016/j.neuron.2018.06.029>
Paul, S. et al, 2020. *Nat Commun*, <https://doi.org/10.1038/s41467-020-15277-x>
Tso, C.F. et al, 2017. *Curr Biol*, <https://doi.org/10.1016/j.cub.2017.02.037>

Reviewer #3 (Remarks to the Author):

It has been hypothesized that the daily timing of glucocorticoids release is controlled by SCN neuron outputs. There is ample evidence supporting the importance of SCN neurons in control of circadian rhythms in glucocorticoids release, but the circuitry mechanisms are not yet clearly elucidated. This study provides valuable insights. By employing long-term fiber photometry in-vivo recoding, the authors of this study observed that PVNCRH neurons exhibit synchronous daily rhythms in *Per2* gene expression and neuron activity. These recording results are novel and exciting, not only because it provides first in vivo circadian rhythms data of PVNCRH neurons, but also because it links neuron activity outside of the SCN with daily rhythms in corticosterone release. Moreover, they found that selective ablation clock gene *Bmal1* in CRH neurons blunts circadian rhythms in corticosterone, which convincingly demonstrate that the PVNCRH clock is essential for corticosterone rhythmicity.

Consistent with recent findings that activation/inactivation of SCN VIP neurons affected corticosterone release (Paul et al 2020), this manuscript examined the impact of SCN VIP neuron activity on corticosterone rhythmicity and levels in more details. The authors went further to show

that the SCNVIP neurons provide inhibitory inputs to PVNCRH neurons and entrain PVN clock gene rhythm. Overall, the authors provide a comprehensive and compelling set of data. Together, the findings from Jeff R. Jones et al. convincingly demonstrated the corticosterone rhythmicity is dependent on gene expression and neuronal activity rhythms in both SCNVIP and PVNCRH neurons, significantly advanced our understanding about how signals from SCN neurons can be transmitted to downstream PVNCRH neurons to generate circadian rhythms in corticosterone release.

While overall a novel and interesting study, I do have some concerns awaiting to be addressed.

- 1) Ablation of the *Bmal1* in CRH neurons reduces the amplitude and precision of daily corticosterone. However, the authors did not show the impacts of this ablation on the rhythms of clock gene (e.g., *Per2*) and calcium activity or electrophysiological characteristics in PVNCRH neurons. Without this data, it is difficult to know exactly what changes in PVNCRH neurons blunts corticosterone rhythms.
- 2) The authors showed that axons of SCNVIP neurons terminate close to PVNCRH neurons. They also found that activation of SCNVIP neurons inhibits most PVNCRH neurons, whereas inhibition of SCNVIP neurons increases PVNCRH neuronal activity. Are there synaptic connections between SCNVIP and PVNCRH neurons? Monosynaptic or polysynaptic? Monosynaptic retrograde tracing experiments will be helpful to clarify this issue.
- 3) There are several populations of CRH neurons in the brain. So *BMAL1* was also ablated in other CRH neurons of *CrhCre/Cre; Bmal1^{fl/fl}* mice. The authors should mention this, at least in the discussion, although *BMAL1* KO in other CRH neurons is unlikely to have an effect on the corticosterone rhythmicity.

Minor concerns:

1. Should Fig. 1C be labelled ZT/CT?

We have changed clock time to ZT/CT in **Fig. 1c**.

2. While we appreciate that the authors are measuring Venus fluorescence (yellow), will yellow be hard to see in the final figure?

We replaced the yellow color for Venus with bright green.

3. The same symbols should be used in Fig1F and G.

We have changed the symbols in **Figs. 1f and g**.

4. Another minor point is that the authors use yellow to indicate yellow fluorescence in Fig. 1 but also use yellow bars to indicate day. Maybe just use white bars for day?

Because we changed the color of Venus, we decided to keep yellow to indicate day.

5. In Fig 4C, the effect that we see is a suppression of amplitude not the phase shift.

The phases in **Fig. 4** have been reanalyzed as per R2. We have revised **Fig. 4** and added **Fig. S4b** to show the change in amplitude but not phase.

6. In Fig. 4D, the blue line looks to be in the wrong place. Also, we are not sure of the format of 4D, it seems to be a messy way of showing the phase shifts so clearly illustrated in the polar plot.

The previous **Fig. 4d** phase plot has been removed.

7. Could the authors add the peak/trough plot to Fig. 4?

We thank the reviewer for the suggestion to include peak/trough plots. These plots have been added throughout the manuscript (**new Figs. 3g, 4d, 4f**) and recalculated for **new Fig. 5d** (ratio was accidentally inverted in the initial submission).

8. Again in Fig4F, we have a hard time seeing the phase shift. The polar plots are clear.

The phases in **Fig. 4** have been reanalyzed as per R2. There is still a significant phase shift post-CT 0 stimulation (**previous Fig. 4f**) as analyzed in **new Fig. S4d**. We have moved the Rayleigh plots for these figures to the supplemental.

9. In Fig5, it appears that the increase in amplitude and the shift in phase is largely driven by 2-3 mice. Can the authors provide some explanation? Did these mice drink more water and get a higher dose of CNO?

We thank the reviewer for this important comment. The phases in **Fig. 5** have been reanalyzed as per R2. We have replaced the Cosine-calculated amplitude data (**previous Fig. S5b**) with peak finding methods to present peak-trough ratios (**new Fig. 5d**) and phase plots (**new Fig. 5e**). These analyses revealed more similarity between mice of the same genotype and treatment (Two-Way RM ANOVA, Subject F(14,28) = 0.98, p = 0.49). We have added more information to the results (Lines 645-647)

indicating that viral expression was similar between mice, and the total amount of CNO water consumed by each mouse was similar (5-8 ml, no difference between control and experimental mice).

R2

Major concerns:

1. In Figure 2h-j the authors show ex vivo data of calcium recordings from semi-acute PVN slices, claiming a phase coherence between cells in the slice, but can only record rhythms for approximately 36h due to the rapid damping of oscillations. In the first 24 hours of a recording like this, there will be cellular synchronization by exogenous experimental factors. The analysis of phase coherence should be conducted once these effects have had time to dissipate to be able to comment on this aspect of network time keeping. In the current recording configuration, obviously this is not possible due to the rate of PVN damping. An experiment like this would be more convincing had it been done on the SCN co-cultures used in Figure 6 which persist for up to a week.

For the past three months, we attempted these challenging long-term, in vitro fluorescence imaging experiments, but found that imaging of cellular calcium in CRH neurons lasted reliably for about 36 h before fading into the background autofluorescence on our epifluorescence microscope. We include new bioluminescence data (**new Figs. 6i, 6j**) clearly showing how SCN co-cultures sustain daily rhythms in the PVN and stimulation of the SCN can entrain the PVN. We also include additional analysis of GCaMP in the cultured PVN showing that the time of peak Ca^{+2} is independent of the start time of recording (**new Fig. S2k**), providing further evidence that the PVN acts as a damped circadian oscillator entrained and sustained by the SCN.

2. In the experiment where BMAL1 was deleted in the CRH cells of the PVN (Figure 3), was there any loss of BMAL1 in cells elsewhere in the brain aside from the PVN? In the text, it seems to me, that this is presented as a PVN specific deletion. However, CRH is expressed broadly across the brain as revealed by intersectional genetic imaging using the same CRH-Cre mouse line (Peng et al, 2017). Reference to this fact should be made clearer in the text as it will affect the interpretation of this data. The authors should also present the extent to which BMAL1 is deleted in other brain areas, and comment on how this might affect the interpretation of the corticosterone and locomotor activity observations.

We agree with the reviewer that our transgenic *Crh*^{Cre/Cre}; *Bmal1*^{fl/fl} mice have *Bmal1* deleted outside the PVN. We have added immunohistochemistry showing that BMAL1 was absent in the majority of CRH neurons outside the PVN (in, for example, the central amygdala and BNST, **new Fig. S3k**). Importantly, we also find that the normal stress-related increases in activity of CRH neurons in our knockout mice is not significantly different from stress-related increases in activity in WT mice (**new Figs. S3d, e**). Based on the literature, *Bmal1* knockdown in CRH neurons outside the PVN is unlikely to have an effect on CORT rhythms (Ixart et al. 1982; Shor-Posner et al. 1985; Dallman et al. 1989; Dedic et al. 2018; Dedic et al. 2019). We include and interpret this as evidence that loss of *Bmal1* in CRH neurons, most likely in the PVN, results in loss of circadian, but not stress-induced, regulation of corticosterone (Lines 180-182, Lines 412-419).

3. Is genetic ablation of BMAL1 just blunting CRH production resulting in the loss of corticosterone amplitude observed? Have the authors looked at the levels of CRH (at the mRNA and protein levels) in their knockout model? It may be that killing the clock in these cells suppresses the global production of CRH.

We have added CRH staining in PVN from *Crh*^{Cre/Cre}; *Bmal1*^{+/+}, *Crh*^{Cre/+}; *Bmal1*^{+/+}; and *Crh*^{Cre/Cre}; *Bmal1*^{fl/fl} mice (**new Fig. S3c**). We have added the following text to the methods, Lines 710-711 “we

injected colchicine into the lateral ventricles (1 μ l per side, 1 μ g/ μ l) allowed the mice to recover, and perfused them 24 h later at CT 6.” Lines 725-726 “Brains were processed and stained identically (same antibody concentrations, durations, etc.). Lines 727-728 “using identical laser and capture settings for each image that was to be quantitatively compared.” And to the results, Lines 172-176 “there was no significant difference between the number of CRH-immunolabeled neurons in *Crh*-Cre homozygotes, heterozygotes, or *Crh*^{Cre/Cre}; *Bmal1*^{fl/fl} mice. Together, these data demonstrate that the presence of homozygous *Crh*-Cre and the knockout of *Bmal1* in CRH neurons each has no effect on PVN^{CRH} expression.”

4. A more sensitive intervention to show the necessity of the clock in this circuit would be to manipulate the clock without killing it. Deletion of CK1 ϵ TAU which has been used by this lab previously (Tso et al, 2017) would be a better mechanism to see if mistimed TTFL in the CRH neurons can be overridden by the SCN.

We agree with the reviewer that this experiment would further test whether SCN input can “override” CRH rhythms. However, because it would require major financial resources and more than 3 months of breeding (to generate *Crh*^{Cre/Cre}; *Ckl1*^{tau/tau} double-homozygous animals) and an additional 2 or more months of corticosterone collection and processing, we have elected to not manipulate the speed of the CRH neuron clock in the present study.

5. Did the authors test the effect of BMAL1 deletion in the CRH neurons on corticosterone rhythms under an LD cycle. Are these rhythms still disrupted if the SCN cells are receiving a daily retinal input, or is the disruption observed a solely free-running phenomenon? In the discussion, the authors allude to the fact that the relative “weakness” of the BMAL1-deficient condition could be due to an entrainment vs masking mechanism.

We did not collect CORT from our KO mice in LD. However, as per R3 Major Concern #1, we recorded in vivo calcium activity from CRH BMAL1 KO in both LD and DD (**new Figs. 3d, e, S3f, S3g**). In LD the KO calcium activity rhythm was rhythmic and had the same phase as WT calcium activity. However, we found that waveform of KO PVN CRH calcium activity (**new Fig. S3h**) changed such that these neurons responded acutely to light but failed to continue firing into the early night compared to WT. In DD, KO calcium activity was arrhythmic. These results are consistent with light-induced masking and a lack of photoentrainment of CRH neuron activity. As such, we predict that the CORT rhythm in our KO mice would not be disrupted in LD.

6. Alternatively, have the authors tested optogenetic VIP-cellular activation in a CRH-BMAL1 deficient mouse using their Flp/Cre line to try and discriminate between these entrainment and masking phenotypes? This further level of analysis would elevate these findings.

We agree with the reviewer that this could test whether disrupted CORT rhythms in PVN CRH *Bmal1* knockouts can be rescued by SCN VIP activation. However, we decided that this experiment is beyond the scope of the present study as it would require several additional months of breeding (to generate *Vip*^{Flp/+}; *Crh*^{Cre/Cre}; *Bmal1*^{fl/fl} triple-transgenic animals), virus expression and testing, and corticosterone collection and processing.

7. In Figure 3, the authors observe a 1 hour advance in the corticosterone peak in wild-type mice on successive days, but did not observe the same 1 hour advance in wild-type mice in other experiments presented in Figure 4e and h, and in Figure 5e. Is this something specific to the mixed wild-type cohort used here (in which case, they should be separated) or is it due to some inherent inaccuracy in the sampling method?

We have reanalyzed the phases of all the CORT rhythms (Figs. 3, 4, 5) as per R2's Major Concern #8 (see below). We no longer observe a significant hour advance in the WT cohort in **Fig. 3h** (Two-Way Circular ANOVA by Day $X^2 = 4.22$, $p > 0.05$).

8. In the optogenetic experiments presented in Figure 4, the authors claim to observe phase shifts in the peak of corticosterone in the 3rd day following stimulation of the SCN-VIP cells at CT11-13. I struggle to see this in the traces presented (Figure 4c) - on the 3rd day, the peak of the mean aggregate trace of ChR2 animals is aligned with the mean aggregate trace of EGFP animals. This largely chimes, by eye, with the fits to the data presented in Supplementary Figure 4c. The same also follows for my interpretation of the raw and aggregate data presented for the optogenetic stimulation at CT23-1 (Figure 4f and Supplementary Figure 4g) and the chemogenetics (Figure 5c and Supplementary 5c). In the methods, it is stated that the corticosterone oscillations are fit using a cosinor approach. If the phases are taken from these fits they will invariably have an error term associated with them which will affect the precision of the determined peak. It would be better if the authors determined the peak in the corticosterone from the raw data rather than from these fits which will have some associated inherent error. Due to the inconsistency in what is presented and what is interpreted, I have trouble seeing where the reported phase shifts come from?

We thank the reviewer for the suggestion to reanalyze the CORT peak times from the raw data instead of Cosinor fits. We have removed the Cosinor fits from the manuscript. We have now calculated the peak times as the time of maximum CORT on each day of collection. When two timepoints differ in their CORT levels by <15%, we calculated the CORT peak as the midpoint between those two timepoints. This new analysis gives us a resolution of 2 h that we can use to recalculate phase shifts in **new Figs. 3h, S4b, S4d, and 5e**. Notably, the interpretation of our data largely stays the same. In **new Fig. 3h**, *Crh*^{Cre/Cre}; *Bmal1*^{fl/fl} mice still have randomly-distributed CORT peaks on each day of collection (vs. WT that peak at around the same time each day). In **new Fig. 5e**, hM4Di-expressing mice treated with CNO on Day 2 still exhibit a significant phase delay compared to EGFP-expressing mice. In **new Fig. S4d**, ChR2 mice still exhibit a significant phase delay post-CT 0 stimulation. In **new Fig. S4b**, however, we find that ChR2 mice no longer exhibit a significant phase delay post-CT 12 stimulation when phases are calculated with this method. We have revised the text to reflect these new analyses.

9. Additionally, the authors report phase shifts of between 90 minutes and 2 hours: how have they come to these numbers? The phases are calculated on a single cycle based on data where the sampling rate, and therefore resolution, is 4 hours. Phase shifts of these magnitudes seem non-sensical.

CORT peak times have been recalculated as mentioned in R2 Major Concern #8.

10. For the CT11-13 optogenetic data, the authors state in the discussion that the locomotor activity rhythms and corticosterone rhythms dissociate based on locomotor activity presented in a previous paper (Mazuski et al, 2018) and the corticosterone data presented here. However, in Supplementary Figure 4d the authors present locomotor activity rhythms (actograms) for the same mice that were sampled for the corticosterone monitoring experiment: if it is the case that these two rhythms dissociate why not use this data to present phase differences on an animal-by-animal basis? The authors should not be drawing conclusions based on different experimental cohorts, especially when they should have the relevant data to hand, presented within a Supplementary Figure in the manuscript. This data is presented for the CT23-1 optogenetics and chemogenetics (Supplementary Figures 4j and 5f respectively), but not for this critical time point where the authors speculate on this dissociation. Inclusion and analysis of this data would strengthen this interpretation and provide a new, novel aspect to this work.

We agree with the reviewer that including CT 12 locomotor activity would strengthen the claim that there is a dissociation between locomotor activity and CORT rhythms. We unfortunately did not obtain these data because locomotor activity monitoring was only added to our CORT collectors for a subset of

CT 12-stimulated mice. Because of this, we have removed our locomotor activity data and our claims of a dissociation between locomotor activity and CORT from the updated version of the manuscript.

11. Further to this previous point, from the optogenetics at CT23-1 the authors state that there was no effect of stimulation to shift the onset of locomotor activity, but that the corticosterone peak was delayed by approximately 2 hours. They then state that there is no dissociation between the activity rhythms and corticosterone at this time point as the correlation between the two remains the same. Surely, if the peak of corticosterone has moved independent of the activity onset, then there must have been a dissociation between the two?

We agree with the reviewer and predict that this discrepancy is due to the inherent sampling rate limitation of our CORT analysis. We have removed our locomotor activity data and our claims of a dissociation between locomotor activity and CORT from the updated version of the manuscript.

12. In the ex vivo work presented in Figure 6i-j, the authors state that recordings of PER2::LUC bioluminescence in the SCN were made in parallel to the recordings made in the PVN. This data is not presented. In my opinion, it would strengthen the analysis of the findings if the authors were able to determine any relative phase or period difference between the SCN and the PVN, as well as commenting on any potential lags in phase shifting between the SCN and the PVN - does the PVN immediately follow the SCN phase, or is there a lag between the two?

As mentioned in R2 Major Concern #13, our ChR2 PVN slices in **previous Figs. 6i,j** have a shorter PER2 period than our WT PVN slices. We have redone this experiment with a new set of SCN-PVN co-cultures that have similar periods and initial phases. We now present raw and detrended PER2 traces for the PVN (**new Fig. 6i**) and SCN (**new Fig. S6h**) and present the peak times of ChR2 and WT PVN and SCN in **new Fig. 6j**. We do not observe any significant phase or period difference between the SCN and PVN. There is a phase delay of ~1 h between the ChR2 PVN and SCN after VIP stimulation, but this phase delay is not statistically significant.

13. In the entrainment conditions it looks, by eye, in both the example trace (i) and grouped data (j) that the ChR2 condition has a shorter period than the non-ChR2 controls that is accounting for the relative difference in entrainment. This is evident in the fact that control lines appear to be diverging from the ChR2 lines from the 2nd day, before ChR2 entrainment begins. Have the authors analyzed or corrected the data for potential differences in the periods of these cohorts.

We have redone the experiments and analysis in **previous Figs. 6i-j** with ChR2 and WT slices that have similar periods and initial phases.

Minor concerns:

1. In the data presented in Figure 1, did the authors record behavior simultaneous to the fiber photometry for PVN-CRH neurons? If so, out of interest, how did the period of locomotor activity relate to the period of the PVN-CRH neurons, what was the relative phasing between the two rhythms and did this phase relationship change with different lighting conditions?

We did not simultaneously record locomotor activity rhythms for all mice. We have added **new Fig. S1b** to show a representative LD and DD trace from an animal in which we recorded both Per2-Venus and locomotor activity (via infrared). In general, the period of Per2-Venus and locomotor activity are similar, but the peak of Per2-Venus and acrophase of locomotor activity are separated by around 12 h.

2. Line 291: "isolated PVN cultures rapidly damped (n=5, data not shown)". Why not show this data in the Supplementary Figures? This would strengthen the case that the SCN is entraining rhythmicity within the PVN in the SCN-PVN slices.

We have included this in **new Fig. S6g**.

3. Line 360-361: "SCNVIP neurons greatly increase their activity in response to light at night (Jones et al, 2018), but PVNCRH neurons do not (n=3, data not shown)." What is the rationale for not showing this data? The data previously presented (here and in Paul et al, 2020) predicts that increased activity in the SCN-VIP neurons will acutely decrease activity in the PVN-CRH neurons, so why is it surprising that the PVN-CRH neurons do not increase as well?

We thank the reviewer for this comment and have included these data in **new Fig. S2I**. These data are perhaps not surprising considering that an increase in SCN VIP activity due to light could decrease PVN CRH activity. However, at the same time, the presentation of a light pulse to a mouse in constant darkness is a stressor that could independently activate PVN CRH neurons. With the reviewer's encouragement, these data are included to demonstrate that even though they each have clocks and are needed for normal CORT rhythms, SCN VIP neurons and PVN CRH neurons are each fundamentally different neurons that participate in different circuits (light response vs. stress response).

4. In the discussion, the section relating to the effects of PVN-CRH BMAL1 ablation (lines 367-395) neglects to discuss the effects of BMAL1 ablation outside of the PVN in other CRH neurons and whether BMAL1 ablation may affect the expression of CRH.

We now discuss the effects of BMAL1 ablation outside the PVN and BMAL1 ablation on CRH expression in Lines 412-419. We include data showing that BMAL1 ablation does not affect CRH expression in **new Fig. S3c**.

5. Relating to supplementary Figure 3: The labeling of the colors is difficult to discern in the immunohistochemistry (a). The range of the axes cuts off the peak in some of the WT data (c). The statistics occlude the data points (e). There is no description of the dashed red lines in the corticosterone fits (f). In the legend, μM is used in place of μm .

We have changed the labels in **Fig. S3a**. We have changed the range of the axes in **Fig. S3i (previous Fig. S3c)**. We have moved the statistics away from the data points in **new Fig. S3j (previous Fig. S3e)**. We have removed Fig. S3f and associated Cosine fits from the manuscript. We have changed the **Fig. S3** legend.

6. Relating to supplementary Figure 4: The range of the axes cut off the peak in some of the EGFP corticosterone data (b). There is no description of the dashed red lines in the corticosterone fits (c). The significance bracket is offset from the data (j).

We have changed the range of the axes in **Fig. S4a**. We have removed Figs. S4c and S4j.

7. In the methods, I could not find reference to the breeding strategy or fiber luminometry methodology used to generate the in vivo bioluminescence data presented in Supplementary Figure 1c.

We have added this to the Methods (Lines 484-485, 511-513, 535-536, 557-560).

8. Details of the experimental approach (including the AAV vector) to perform the chemogenetic inhibition experiments in Figure 6 are not present. What AAV vectors were used? What was the genetic background of the animals from which slices were made?

We have added this to the Methods (Lines 491-497, 503-508).

9. In Figure 3, the wild-type cohort are made of a mix of genetic backgrounds. The specific breakdown of the strains/numbers contributing should be listed.

We have added this to the Methods (Lines 488-489).

10. The legends to the majority of figures do not contain sufficient detail and are not clear enough to allow a non-expert in the circadian field to interpret the figures properly. There is a lot of description and interpretation of the data within the legends – this should be left to the main text. The legends are missing vital information to aid with the reader's own interpretation in isolation e.g. the definition of error bars and other figure components etc - a stated requirement of manuscript format.

We have added details to the figure legends to allow a non-expert to interpret the figures. We have also removed our data interpretation from the figure legends and have added information including definitions of error bars to the figure legends.

11. Largely the statistical tests are appropriate to the data used. However, the tests presented in Figure 6e and 6h are, in my opinion, inappropriate. In these tests, the individual cells are treated as independent biological replicates - this is incorrect. These individual cells came from 3 biologically independent mice: the test should reflect this fact. A better approach would be to use a model where the relative contributions of individual cells can be taken into consideration, but where they are not treated as biological replicates for example a linear mixed model approach.

We thank the reviewer for finding our error in statistical analysis. We have reanalyzed **new Figs. 6e and 6h** using two methods. First, we used a nested one-way ANOVA to rule out significant variability between slices as in (Green NH et al. Photoperiod programs dorsal raphe serotonergic neurons and affective behaviors. Curr Biol 2015.).

Fig. 6e ChR2 event frequency: $p = 0.1397$, $F(1, 6) = 2.896$

Fig. 6e ChR2 integrated calcium: $p = 0.4026$, $F(1, 6) = 0.8109$

Fig. 6h hM4Di event frequency: $p = 0.6733$, $F(1, 6) = 0.2062$

Fig. 6h hM4Di integrated calcium: $p = 0.9764$, $F(1, 6) = 0.009$

We next used a Two-Way Repeated Measures ANOVA to compare the cells in each slice before and after treatment.

Fig. 6e ChR2 event frequency, Pre vs. Post: $p < 0.001$, $F(1, 101) = 66.52$, $p < 0.001$

Fig. 6e ChR2 integrated calcium, Pre vs. Post: $p < 0.001$, $F(1, 101) = 57.21$, $p < 0.001$

Fig. 6h hM4Di event frequency, Pre vs. Post: $p > 0.05$, $F(1, 160) = 0.89$, $p > 0.05$

Fig. 6h hM4Di integrated calcium, Pre vs. Post: $p < 0.001$, $F(1, 160) = 55.20$, $p < 0.001$

We have also updated the graphs to depict the mean \pm SEM of the cells in each slice before and after treatment.

12. In Supplementary Table 1, the statistical results for some of factors are missing from the table.

We have corrected and updated the statistical results table.

Major concerns:

- 1) Ablation of the *Bmal1* in CRH neurons reduces the amplitude and precision of daily corticosterone. However, the authors did not show the impacts of this ablation on the rhythms of clock gene (e.g., *Per2*) and calcium activity or electrophysiological characteristics in PVNCRH neurons. Without this data, it is difficult to know exactly what changes in PVNCRH neurons blunts corticosterone rhythms.

We thank the reviewer for this important suggestion. We recorded *in vivo* calcium activity from the PVN of *Crh*^{Cre/Cre}; *Bmal1*^{fl/fl} mice in both LD and DD (**new Figs. 3d, 3e, S3f, S3g**). We found that in LD, *Crh*^{Cre/Cre}; *Bmal1*^{fl/fl} calcium activity was rhythmic and peaked at the same time as CRH WT calcium activity. However, in DD, *Crh*^{Cre/Cre}; *Bmal1*^{fl/fl} calcium activity was arrhythmic. Based on these data, we conclude that a disrupted CRH neuron activity rhythm results in arrhythmic CORT release in our KO mice.

- 2) The authors showed that axons of SCN VIP neurons terminate close to PVNCRH neurons. They also found that activation of SCN VIP neurons inhibits most PVNCRH neurons, whereas inhibition of SCN VIP neurons increases PVNCRH neuronal activity. Are there synaptic connections between SCN VIP and PVNCRH neurons? Monosynaptic or polysynaptic? Monosynaptic retrograde tracing experiments will be helpful to clarify this issue.

We thank the reviewer for this interesting suggestion and have performed three experiments to help clarify the connection between SCN VIP and PVN CRH neurons.

First (**new Fig. S6a, b**), we injected the PVN of *n* = 4 *Vip*-IRES-Cre mice with a retrograde AAV encoding a “CreSwitch” virus (Saunders A, Johnson CA, and Sabatini BL. Novel recombinant adeno-associated viruses for Cre activated and inactivated transgene expression in neurons. *Front Neural Circuits* 2012) that encodes tdTom in the absence of Cre and EGFP in the presence of Cre. After transfection, starter neurons in the PVN and Cre- projection neurons in the SCN would express tdTom, and *Vip*-Cre+ projection neurons in the SCN would express EGFP. We found that a small number of SCN VIP neurons project monosynaptically to the PVN. A subset of EGFP+ processes terminated near cells that co-expressed tdTom (virus starter cells) and CRH.

Second (**new Fig. S6c, d**), we injected the SCN of *n* = 4 *Vip*-IRES-Cre; Ai14 (floxed tdTom) mice with an AAV encoding Cre-dependent, EGFP-tagged synaptophysin. tdTom+ VIP neuron cell bodies and axons were present throughout the ventral SCN. VIP neuron EGFP+ synapses were abundant and present throughout the dorsal-ventral extent of the SCN. In the PVN, however, we only observed a small number of VIP neuron EGFP+ synapses onto PVN CRH neurons.

Third (**new Fig. S6e**) we performed double-label immunohistochemistry for the VIP receptor VPAC2 and CRH in *n* = 4 PVN slices. We found that consistent with prior reports, there was dense VPAC2 labeling throughout the SCN and PVN (An S, et al. Spatiotemporal distribution of vasoactive intestinal polypeptide receptor 2 in mouse suprachiasmatic nucleus. *J Comp Neurol* 2012; Sunkin SM et al. *Allen Brain Atlas: an integrated spatio-temporal portal for exploring the central nervous system. Nucleic Acids Res* 2013). Due to the limitations of the anti-VPAC2 antibody, we and others have found it difficult to clearly resolve the co-expression of VPAC2 and other cell types. However, in our PVN slices, the extent of VPAC2 labeling overlapped that of PVN CRH neuron staining.

Together, these data, combined with our identification of several VIP+ axons terminating near CRH+ neurons (**Fig. 6b, new Fig. S6f**), suggest that monosynaptic connections between SCN VIP and PVN CRH neurons are rare. The dense VPAC2 staining in the PVN suggests that SCN VIP axons that terminate in the PVN volumetrically release VIP to communicate with PVN CRH neurons. Additionally,

SCN VIP neurons may communicate polysynaptically with PVN CRH neurons, perhaps first synapsing in the subparaventricular zone similar to SCN AVP neurons.

- 3) There are several populations of CRH neurons in the brain. So BMAL1 was also ablated in other CRH neurons of *Crh*^{Cre/Cre}; *Bmal1*^{fl/fl} mice. The authors should mention this, at least in the discussion, although BMAL1 KO in other CRH neurons is unlikely to have an effect on the corticosterone rhythmicity.

We agree with the reviewer that our transgenic *Crh*^{Cre/Cre}; *Bmal1*^{fl/fl} mice have *Bmal1* knocked down outside the PVN. We found that BMAL1 was absent in the majority of CRH neurons outside the PVN (in, for example, the central amygdala and BNST, **new Fig. S3k**). We comment on how this might affect the interpretation of our results in the discussion (Lines 412-419). We also agree with the reviewer that, based on the literature, *Bmal1* knockdown in CRH neurons outside the PVN is unlikely to have an effect on CORT rhythms (Ixart et al. 1982; Shor-Posner et al. 1985; Dallman et al. 1989; Dedic et al. 2018; Dedic et al. 2019).

Reviewer comments, second round:

-

Reviewer #1 (Remarks to the Author):

In this work, the authors clarified a critical circuit from SCN VIP-producing neurons to PVN CRH. They carried out this work for the most part using state-of-the-art in vivo imaging techniques in freely behaving mice. As needed, they supplemented this in vivo analysis with work in brain slices. The writing is clear and scholarly. The figures clearly support the main concepts. This circuit is biologically important and dysfunction is associated with a number of diseases.

The serious revisions to the prior reviewer comments have significantly strengthened this study.

Reviewer #2 (Remarks to the Author):

The authors have adequately answered the majority of my concerns, and have incorporated substantial improvements to the manuscript including new experiments and enhanced analysis.

However, I still have reservations about the reported phase shifting in the inhibitory chemogenetics experiment (Figure 5). Is this phase shift (Figure 5e) driven by the fact that there is an emergent second peak in the projected trough of the CORT rhythm? From the raw (Supplementary Figure 5) and mean (Figure 5c) traces, it would seem that the phase of the main peak of CORT is unaltered at ~CT12-15 compared to the control EGFP animals: indicating that there is no shift in the phase of this peak. What is very clear, however, is that a second peak is now evident at ~CT22-6 as stated by the authors.

In my opinion, this question of phasing would be better analysed if the two peaks were treated independently. This would strengthen the model that circadian activation of VIP-neurons in the SCN inhibits the activity of CRH neurons in the PVN (Figure 6a-h), leading to the suppression of the peak in CORT seen in the optogenetics experiments (Figure 4). My interpretation of the experiment would be that chemogenetic inhibition of the VIP-neurons at the time when VIP neurons are already inactive (CT12-15) does not alter the dynamics of the first CORT peak – therefore it is indistinguishable in phase and peak amplitude from the control animals. However, when the VIP neurons would normally become active between around CT0-8 (Hermansteyne et al, 2016; Jones et al, 2018; Patton et al, 2020; Paul et al, 2020) they are inhibited chemogenetically meaning that they can no longer suppress the PVN-CrH neurons. This leads to a blunting of the trough and gives rise to the secondary peak seen in this experiment and the subsequent reduction of the relative amplitude of the oscillation across 24h. Thus, in this interpretation, these two peaks likely arise from different sources in this experiment – the first peak arises from the normal circadian electrical quiescence of the VIP cells, while the second peak is driven by chemogenetic inhibition of VIP cells: they should therefore be treated independently.

References

Hermansteyne et al (2016). Distinct Firing Properties of Vasoactive Intestinal Peptide-Expressing Neurons in the Suprachiasmatic Nucleus. *J Biol Rhythms* 31(1): 57-67.
Jones et al (2018). SCN VIP Neurons Are Essential for Normal Light-Mediated Resetting of the Circadian System. *J Neurosci* 38(37):7986-7995.
Patton et al (2020). The VIP-VAPC2 neuropeptidergic axis is a cellular pacemaking hub of the suprachiasmatic nucleus circadian circuit. *Nat Comms* 11, 3394.
Paul et al (2020). Output from VIP cells of the mammalian central clock regulates daily physiological rhythms. *Nat Comms* 11, 1453.

Reviewer #3 (Remarks to the Author):

The authors have addressed all my concerns with new experiments and other edits in the revised manuscript.

The authors have adequately answered the majority of my concerns and have incorporated substantial improvements to the manuscript including new experiments and enhanced analysis.

However, I still have reservations about the reported phase shifting in the inhibitory chemogenetics experiment (Figure 5). Is this phase shift (Figure 5e) driven by the fact that there is an emergent second peak in the projected trough of the CORT rhythm? From the raw (Supplementary Figure 5) and mean (Figure 5c) traces, it would seem that the phase of the main peak of CORT is unaltered at ~CT12-15 compared to the control EGFP animals: indicating that there is no shift in the phase of this peak. What is very clear, however, is that a second peak is now evident at ~CT22-6 as stated by the authors. In my opinion, this question of phasing would be better analysed if the two peaks were treated independently. This would strengthen the model that circadian activation of VIP-neurons in the SCN inhibits the activity of CRH neurons in the PVN (Figure 6a-h), leading to the suppression of the peak in CORT seen in the optogenetics experiments (Figure 4).

My interpretation of the experiment would be that chemogenetic inhibition of the VIP-neurons at the time when VIP neurons are already inactive (CT12-15) does not alter the dynamics of the first CORT peak – therefore it is indistinguishable in phase and peak amplitude from the control animals. However, when the VIP neurons would normally become active between around CT0-8 (Hermansteyne et al, 2016; Jones et al, 2018; Patton et al, 2020; Paul et al, 2020) they are inhibited chemogenetically meaning that they can no longer suppress the PVN-CrH neurons. This leads to a blunting of the trough and gives rise to the secondary peak seen in this experiment and the subsequent reduction of the relative amplitude of the oscillation across 24h. Thus, in this interpretation, these two peaks likely arise from different sources in this experiment – the first peak arises from the normal circadian electrical quiescence of the VIP cells, while the second peak is driven by chemogenetic inhibition of VIP cells: they should therefore be treated independently.

We thank the reviewer for this suggestion and agree with their interpretation of the data. The phase shift observed in **previous Fig. 5** was indeed due to the second peak, as we had calculated the phase as the midpoint between the two peaks. We have recalculated the phases on those plots in **new Fig. 5d** to depict the phases of the first and second corticosterone peaks as determined by a peak-finding algorithm as used in **new Supplementary Fig. 5a**. We have also recalculated the peak-trough amplitude ratios in **new Fig. 5e** to account for the peak, instead of mean, corticosterone amplitude during the peak and trough of the corticosterone rhythm. Finally, we have added **new Supplementary Fig. 5b** to show an increase in the trough amplitude (and thus a “second peak”) in the majority of hM4Di compared to EGFP mice.

We have changed the Results section to account for these changes on lines 262-269, and the Discussion section to account for these changes on lines 425-436.